# Fertilization turns a rubber plantation from sink to methane source

Daniel Epron[1,2], Rawiwan Chotiphan[3], Zixiao Wang[1], Ornuma Duangngam[4], Makoto Shibata[5], Sumonta Kumar Paul[1], Takumi Mochidome[1], Jate Sathornkich[2], Wakana A. Azuma[6], Jun Murase[2,7], Yann Nouvellon[4,8], Poonpipope Kasemsap[2,4], and Kannika Sajjaphan[2]

[1]Graduate School of Agriculture, Kyoto University, Kyoto 606-8502, Japan

[2]Faculty of Agriculture, Kasetsart University, Bangkok 10900, Thailand

[3]Sithiporn Kridakara Research Station, Faculty of Agriculture at Kamphaeng Saen, Kasetsart University, Prachuap Khiri khan 77170, Thailand

[4]DORAS Centre, Kasetsart University, Bangkok 10900, Thailand;

[5]Graduate School of Global Environmental Studies, Kyoto University, Kyoto 606-8501, Japan

[6]Graduate School of Agricultural Science, Kobe University, Kobe 657-8501, Japan

[7]Graduate School of Bioagricultural Sciences, Nagoya University, Nagoya 464-8601, Japan

[8]CIRAD, UMR Eco&Sols, 2 Place Viala, 34060 Montpellier Cedex 2, France

*Correspondence to*: Daniel Epron (daniel.epron.3a@kyoto-u.ac.jp); Kannika Sajjaphan (agrkks@ku.ac.th)

**Abstract.** The rapid expansion of rubber cultivation, driven by the demand for natural rubber in the tire industry constitutes a significant land-use change in Southeast Asia. This significant land-use change has reduced soil methane ($CH_4$) uptake, thereby weakening atmospheric $CH_4$ removal over extensive areas. While fertilization is a widespread practice in rubber plantations, its role in further weakening the soil $CH_4$ sink remained poorly understood. Over 1.5 years, we measured soil $CH_4$ fluxes biweekly in an experimental rubber plantation with four distinct fertilization treatments to evaluate their impact

on the soil $CH_4$ uptake. Our findings revealed that fertilization not only reduced soil $CH_4$ consumption, but also increased soil $CH_4$ production. The difference in soil $CH_4$ uptake between unfertilized plots (-2.9 kg $CH_4$ ha$^{-1}$ yr$^{-1}$) and those with

rational fertilization (-2.1 kg $CH_4$ $ha^{-1}$ $yr^{-1}$) was moderate. Recommended fertilization rates reduced soil $CH_4$ uptake by 60% (-1.1 kg $CH_4$ $ha^{-1}$ $yr^{-1}$), and heavy fertilization transformed the soil into a net source of $CH_4$ (+0.3 kg $CH_4$ $ha^{-1}$ $yr^{-1}$). The suppression of soil $CH_4$ oxidation was likely driven by increased mineral nitrogen in the soil solution and soil acidification, while elevated dissolved organic carbon likely stimulated $CH_4$ production in the topsoil. Most rubber tree trunks emitted $CH_4$, likely of internal origin. Trunk $CH_4$ fluxes ranged from -0.10 to 0.51 nmol $s^{-1}$ per tree, with no significant fertilization effect. At the national level, adopting rational fertilization practices in Thailand could enhance the net soil $CH_4$ sink by 5.9 Gg $CH_4$ $yr^{-1}$. However, this mitigation strategy would have a limited impact on the overall greenhouse gas budget of the agricultural sector in Southeast Asia, unless it is extended to other tree plantations and cropping systems.

## 1 Introduction

Methane ($CH_4$) is the second most impactful anthropogenic greenhouse gas, contributing approximatively one-third to the anthropogenic radiative forcing (IPCC, 2021). The Global Methane Pledge endorsed by 111 countries at the 26th session of the Conference of the Parties (COP26) to the United Nations Framework Convention on Climate Change, commits for a 30% reduction of emissions from anthropogenic sources by 2030. Atmospheric $CH_4$ removal (negative $CH_4$ emission) may be necessary to achieve this goal (Jackson et al., 2021). Soils serve as the largest biological sink for atmospheric $CH_4$, with an annual global uptake estimated as 25 – 45 Tg (Saunois et al., 2020). Upland tropical forest soils, in particularly, play a critical role in this global sink, providing a valuable ecosystem service.

Southeast Asia has experienced extensive land-use changes over recent decades, with rubber and oil palm cultivation being the dominant agricultural expansion. Rubber plantations now occupy over 142,000 $km^2$ in the region (Wang et al., 2023), and this area is projected to quadruple within the next 30 years, driven by increasing global demand of natural rubber, particularly for tire manufacturing (Fox et al., 2014). While land-use comparisons have been extensively studied, to our knowledge, no previous research has specifically addressed the effect of management practices—particularly fertilization—on the $CH_4$ budget of rubber plantations. A recent study found no effect of reduced fertilization on soil $CH_4$ uptake in an oil palm plantation in Indonesia (Chen et al., 2024).

A recent survey of Thai rubber smallholders, who manage 90% of the country's rubber plantations, revealed that fertilization is nearly ubiquitous. Two-thirds of these plantations employ intensive (N application above 53 kg ha$^{-1}$yr$^{-1}$) or very intensive (N application above 94 kg ha$^{-1}$yr$^{-1}$) fertilization rates, even though the benefit of such practices are not well demonstrated (Chambon et al., 2018). Given the importance of agriculture as the largest anthropogenic $CH_4$ source, mitigation strategies targeting $CH_4$ emission from rice cultivation, enteric fermentation, manure management and residue

burning have been extensively evaluated (Smith et al., 2021). However, the potential of rational fertilization to enhance soil $CH_4$ uptake has not yet to be assessed, although it may be another effective mechanism for atmospheric $CH_4$ removal for agriculture.

    The conversion of forests to rubber plantations in Southeast Asia is known to reduce soil $CH_4$ uptake compared to natural forests (Ishizuka et al., 2002, 2005; Lang et al., 2017, 2019; Werner et al., 2006). Given the current extent and

anticipated expansion of rubber plantations in Southeast Asia and other tropical regions, a weaker soil $CH_4$ sink could have significant implications for the global $CH_4$ budget. The net rate of $CH_4$ uptake, defined as the difference between $CH_4$ production and oxidation rates (Le Mer and Roger, 2001), depends primarily on the air-filled porosity (AFP) of the soil. AFP itself is determined by soil water content (SWC) and total porosity (Epron et al., 2016; Kruse et al., 1996). A high AFP enhances gas diffusion within the soil, thereby promoting microbial $CH_4$ oxidation. It has been hypothesized that the

reduction in soil $CH_4$ uptake following forest conversion is primarily due to increased SWC, attributed to lower water use by rubber trees compared forest trees (Lang et al., 2020). However, studies have reported higher evapotranspiration rates in rubber plantations than in tropical forests (Giambelluca et al., 2016; Guardiola-Claramonte et al., 2008; Niu et al., 2017; Tan et al., 2011), which contradicts the assumption of higher SWC. The underlying causes of reduced soil $CH_4$ uptake in rubber plantations compared to forests remain unclear, particularly the extent to which management practice may mitigate or

exacerbate this weakening of the soil $CH_4$ sink.

    Although fertilization is a common practice in rubber plantations, its effects on soil $CH_4$ uptake have not yet been documented. Fertilization can enhance tree growth, potentially increasing tree water use if the transpiration efficiency—the ratio of dry biomass accumulation per unit water transpired—does not improve significantly. Higher rates of evapotranspiration could lower SWC, particularly in the upper soil layers, thereby increasing AFP. This, in turn, would

facilitate the diffusion of atmospheric $CH_4$ and oxygen ($O_2$) into the soil pores, where $CH_4$ is oxidized by methanotrophs, unless SWC becomes too low, which could limit microbial activity and hinder $CH_4$ oxidation (Borken et al., 2006; Bras et al., 2022; von Fischer et al., 2009; Qiu et al., 2024).

    Fertilizers can also have direct effects, either positive or negative, on soil $CH_4$ uptake. They can alleviate nitrogen (N) or phosphorus (P) limitations for methane oxidizing bacteria (MOB) in tropical forests soils, depending on the nutrient

status of the soil. Like other microorganisms, MOB require N and P to sustain their growth and activity (Bodelier and Laanbroek, 2004; Martinson et al., 2021; Veldkamp et al., 2013). However, excessive nitrogen inputs can reduce soil $CH_4$ oxidation (Lee et al., 2023; Steudler et al., 1989; Zhang et al., 2020). Several mechanisms have been proposed to explain this inhibition. Ammonia-oxidizing bacteria, which can oxidize $CH_4$ instead of ammonium ($NH_4^+$) under low $NH_4^+$ availability due to the similarity between the enzymes ammonia monooxygenase and methane monooxygenase, shift their activity to

$NH_4^+$ oxidation when N limitation is alleviated (Bédard and Knowles, 1989). A similar substrate competition occurs when $NH_4^+$ competes with $CH_4$ for the active site of methane monooxygenase (King and Schnell, 1994; O'Neill and Wilkinson, 1977). However, unlike $CH_4$, $NH_4^+$ does not provide carbon to sustain the growth of methanotrophic bacteria and produces nitrite, which is toxic to them (Schnell and King, 1994). Additionally, cations in fertilizers, such as potassium, can compete with $NH_4^+$ for exchange sites on clay-humus complexes in the soil, releasing $NH_4^+$into the soil solution (King and Schnell,

1998). Nitrate ($NO_3^-$), possibly after been reduced to nitrite ($NO_2^-$), has also been identified as a potent inhibitor of $CH_4$ oxidation in some soils (Mochizuki et al., 2012; Reay and Nedwell, 2004; Wang and Ineson, 2003). Excessive N fertilizer application can further acidify the soil (Qu et al., 2014), which is known to negatively impact soil $CH_4$ oxidation (Benstead and King, 2001; Hütsch et al., 1994). Conversely, phosphate ($PO_4^{3-}$) has been found to mitigate the inhibitory effect of N on $CH_4$ oxidation at certain sites, while at others, it has been suspected of stimulating methanogenesis, thereby reducing net

atmospheric $CH_4$ uptake (Zhang et al., 2011; Zheng et al., 2016).

    In addition to anoxic conditions, the main factor controlling methanogenesis is the availability of organic substrates derived from primary production (Liu et al., 2011; Whiting and Chanton, 1993). This availability can increase with fertilizer inputs, due to greater production of above- or below-ground litter (including sloughed-off cells), enhanced decomposition rates, and increased root exudation (Banger et al., 2012; Hobbie, 2005; Melillo et al., 1982; Zhu et al., 2013). Significant

increase in $CH_4$ emissions have been reported from eutrophied lakes and ponds in agricultural catchments, mangrove sediments receiving sewage discharges or anthropogenic nutrient inputs, and irrigated and fertilized young tree plantations on lowland soils (Allen et al., 2011; Beaulieu et al., 2019; Huttunen et al., 2003; Purvaja and Ramesh, 2001; Rabbai et al., 2024; Sotomayor et al., 1994). In contrast, nitrate additions have been reported to decrease $CH_4$ emissions in rice paddies and wetlands, highlighting the variability in methanogenesis responses depending on environmental conditions and nutrient dynamics (Kim et al., 2015; Roy and Conrad, 1999).

Trees can both emit and uptake $CH_4$, complicating our understanding of the greenhouse gas budget of forest ecosystems and tree plantations (Barba et al., 2019b; Gauci et al., 2024). For example, trees contribute up to 87% of the total ecosystem $CH_4$ flux in a Southeast Asian tropical peat forest (Pangala et al., 2013). While $CH_4$ produced in the soil or sediment is a well-recognized sources of tree $CH_4$ emissions in forested wetland (Gauci et al., 2010; Sakabe et al., 2021; Terazawa et al., 2015), $CH_4$ can also be produced endogenously within the heartwood under anaerobic conditions (Epron et al., 2023; Pitz et al., 2018; Wang et al., 2017). Regardless of whether $CH_4$ originates from the soil or is produced within the tree, it can be further consumed by methanotrophic bacteria living in the stem bark; these MOB can also oxidize atmospheric $CH_4$ (Gauci et al., 2024; Jeffrey et al., 2021; Machacova et al., 2021).

In this study, we measured soil $CH_4$ fluxes over one and a half years at two-week intervals in an experimental rubber plantation with four fertilization treatments applied over eight years. The objective was to assess the impact of fertilizers on the soil $CH_4$ uptake. Specifically, we hypothesized that fertilization decreases soil $CH_4$ oxidation, while also considering the possibility that fertilization could stimulate $CH_4$ production, particularly during the rainy season. To better understand the factors driving changes in soil $CH_4$ uptake in response to fertilization, we also monitored soil $CH_4$ concentration gradients, mineral N and $PO_4^{3-}$ availability using ion exchange resin bags, as well as dissolved organic carbon (DOC), total dissolved nitrogen (TDN), and other edaphic factors. Additionally, we measured $CH_4$ emissions from the tree trunk surface to assess the extent to which they offset soil $CH_4$ uptake or contributed to the combined net $CH_4$ emissions from trunks and soil.

## 2 Materials and Methods

### 2.1 Experimental site

The experimental rubber plantation is located at the Sithiporn Kridakara Research Station of Kasetsart University in Prachuap Khirikhan province, Thailand (10°59'13"N, 99°29'22"E, 10 m a.s.l.). The site lies at the transition between two climate groups according to the Köppen climate classification: tropical rain forest (Af) and tropical monsoon climate (Am). Annual rainfall averaged 1,700 mm between 2010 and 2023, with a wet season extending from May to November and a dry season from December to April. October and November are the wettest months, receiving over 250 mm of rain per month on average. The deep sandy loam soil is classified as Arenic Kandiudults (Soil Survey Staff, 2022) or Ferralic Chromic Acrisols (Loamic, Geric, Ochric) (IUSS Working Group WRB, 2022), developed on Cenozoic sedimentary rocks. The rubber plantation (9 ha, clone RRIM600) was established in 2007 replacing a coconut plantation at a planting density of 500 trees ha$^{-1}$, in accordance with the recommendation of the Rubber Research Institute of Thailand. Latex harvesting by taping the bark of the trees began in May 2014 and continues annually from May to February (Chotiphan et al., 2019).

A complete randomized block design was implemented with four blocks and four fertilizer treatments (N/P/K): T1 (no fertilizer), T2 (37/22/50 kg ha$^{-1}$ yr$^{-1}$), T3 (90/40/85 kg ha$^{-1}$ yr$^{-1}$), and T4 (153/68/144 kg ha$^{-1}$ yr$^{-1}$). Fertilization treatments began in May 2014, coinciding the start of latex harvesting by tapping. Treatment T2 represents a rational fertilization level recommended by agronomists specializing in rubber cultivation (Gohet et al., 2013). Treatment T3 falls within the range of rates recommended by Thai public institutions for mature rubber plantations, though 40% of rubber farmers exceed these recommendations (Chambon et al., 2018), a practice represented by treatment T4. Fertilizer for T2 was applied only during the early rainy season (May) while a second application was made during the late rainy season (October) for T3 and T4. Fertilizer was applied by broadcasting, with workers walking along the interrow at approximately 2 m from the planting rows. The 16 elementary plots (four treatments across four blocks) each contained 108 trees and covered an area of 2160 m².

## 2.2 Methane flux measurement

Soil $CH_4$ fluxes ($F_{S\text{-}CH4}$) were measured over one and a half years at approximately two-week intervals (37 measurement dates between September 6, 2022 and February,19 2024). A total of 96 PVC collars (20 cm in diameter and 13 cm in height), inserted 6 cm into the soil, were distributed across four blocks and four fertilizer treatments. Each plot contained six collars, positioned at three distances from the tree rows (0.7, 2.0, and 3.3 m) to capture spatial variability associated with the planting scheme and fertilizer application. The collars were covered with a 20 cm soil chamber (Li 8100-103, Li-Cor; Lincoln, USA), and change in the $CH_4$ mole fraction inside the closed chamber was recorded for 3 minutes at a frequency of 1 Hz using a cavity-enhanced absorption spectroscopy gas analyser (Li 7810). Soil temperature at a depth of 10 cm ($T_{SOIL}$) and volumetric soil water content (SWC) in the 0–6 cm layer were measured simultaneously near each collar. $T_{SOIL}$ and SWC measurements were performed using a digital thermometer and a soil moisture probe (SM150, Delta-T Devices, Cambridge, UK).

Trunk $CH_4$ fluxes ($F_{T\text{-}CH4}$) were measured in August 2023, October 2023, and February 2024 on 8 to 13 trees per treatment. Rectangular polypropylene chamber bases (80 $cm^2$) were affixed to the bark surface with neutral seal putty after gently brushing the bark was to ensure proper adhesion. Chambers were closed during measurement by attaching a polypropylene lid lined with a silicone rubber gasket and connected to the gas analyser. Measurements were performed first at 40-60 cm above the ground. If the increase in the $CH_4$ mole fraction exceeded 0.01 ppb $s^{-1}$, additional measurements were taken at 150-170 cm and, if necessary, at 190-220 cm following the same decision rule. $F_{T\text{-}CH4}$ values were scaled to the tree level (nmol $CH_4$ $s^{-1}$ $tree^{-1}$) by multiplying flux measurements by the corresponding stem surface areas. The trunk of each tree was divided into virtual segments, for which both $F_{T\text{-}CH4}$ and diameter were measured at the chamber location. The length of each virtual segment was calculated as the difference between half the distance to the chamber located above (or 3.5 m height for the upper chamber) and half the distance to the chamber located below (or the height above the ground for the lower chamber). The surface area of each segment was calculated assuming a cylindrical shape and then multiplied by the flux per unit area measured at the corresponding chamber. The integrated fluxes of all trunk segments were summed for each individual tree. Finally, $F_{T\text{-}CH4}$ was multiplied by tree density to expressed $F_{T\text{-}CH4}$ at the plantation scale, allowing comparison with $F_{S\text{-}CH4}$ on a soil surface area basis.

The slopes of the linear variations in $CH_4$ mole fractions over time were used to calculate $CH_4$ flux, discarding the first 60 s of measurements (Epron et al., 2023; Plain et al., 2019):

$$F_{CH_4} = \frac{\Delta[CH_4]}{\Delta t} \frac{V \times P_{atm}}{A \times R \times (T_{air} + 273.15)} \qquad [1]$$

where $F_{CH4}$ is the net $CH_4$ flux (nmol $m^{-2}$ $s^{-1}$) from either soil or trunk, $\frac{\Delta[CH_4]}{\Delta t}$ is the slope of linear $CH_4$ mole fractions variations over time (ppb $s^{-1}$), $V$ is the system volume ($m^3$), including the chamber, part of the collar protruding from the soil, tubing, and analyser, $A$ is the soil or trunk surface area covered by the chamber ($m^2$), $T_{air}$ is the air temperature (°C), $R$ is the ideal gas constant (8.314 J $K^{-1}$ $mol^{-1}$), and $P_{atm}$ is the atmospheric pressure, assumed constant at 101,325 Pa. Based on the manufacturer's specifications (precision of 0.60 ppb $CH_4$ at 2 ppm with 1-second averaging), the minimal detectable flux was estimated at 0.005 nmol $m^{-2}$ $s^{-1}$ for soil and 0.003 nmol $m^{-2}$ $s^{-1}$ for trunks (Bréchet et al., 2021; Epron et al., 2023). Positive $CH_4$ fluxes indicate net emission to the atmosphere, while negative fluxes represent net uptake.

Cumulative annual soil $CH_4$ fluxes were calculated for each collar using linear interpolations of $F_{S-CH4}$ between consecutive measurement date following the method described by Gana et al. (Gana et al., 2018) for $CO_2$ fluxes. Results were expressed in kg $CH_4$ $ha^{-1}$ $yr^{-1}$ and calculated for two periods: September 6, 2022 and September 5, 2023, and February 20, 2023 and February 19, 2024. These two periods overlap by approximately 6 months due to the late start of the project caused by international travel restrictions during the Covid-19 pandemic in Japan and Thailand until summer 2022. Nevertheless, the first one-year period was wetter than the second, with cumulative rainfall of 1,889 mm and 1,565 mm, respectively.

**2.3 Soil methane mole fraction**

Soil $CH_4$ mole fractions ($[CH_4]_S$) were measured only three times during the study, at two soil depths (10 and 40 cm) near 24 soil collars (six per fertilization treatments, though not evenly distributed across the four blocks). In August 2023, two stainless-steel pipes (inner diameter: 5 mm), 20 and 50 cm in length, were vertically inserted into the soil next to each other, with a 10 cm gap between them. The buried ends of the pipes were pinched closed, and two side holes (2 mm in diameter)

were drilled just above the closed end. The opposite ends of the pipes protruded 10 cm above the soil surface and were sealed with septa.

One week later, an air sample (0.5 ml) was drawn from each pipe using a syringe through the septum and injected into the sample kit (Li 7800-110), which was connected to the gas analyser. Before injecting, the sample kit and analyser loop were flushed with ambient air and closed. The mole fraction of $CH_4$ in the closed loop was recorded for 1 min before injection and for 2 min after injection. The mole fraction of $CH_4$ in the injected air sample was calculated as follows:

$$[CH_4]_S = \frac{V_L \times ([CH_4]_P - [CH_4]_L) + V_S \times [CH_4]_P}{V_S} \qquad [2]$$

where $V_L$ and $V_S$ are the volumes of the loop and injected sample, respectively. The indices for $[CH_4]$ indicate the mole fractions in the loop before injection ($L$), in the loop after injection ($P$), and in the air sample ($S$). The same sampling procedure was repeated in October 2023 and February 2024.

Gradients in $CH_4$ mole fraction within the two soil layers (0 -10 cm and 10- 40 cm) were calculated as the difference between $[CH_4]_S$ between the upper and lower depths of each layer, divided by the depth difference (d):

$$\Delta CH_4 = \frac{\left([CH_4]_{lower} - [CH_4]_{upper}\right)}{d_{lower} - d_{upper}} \qquad [3]$$

The $CH_4$ mole fraction in the ambient air, measured 15 cm above the ground before closing the loop, was used as the reference value at 0 cm depth. A negative $\Delta CH_4$ indicated that $CH_4$ oxidation dominated over $CH_4$ production (net $CH_4$ consumption), while a positive $\Delta CH_4$ value indicated that $CH_4$ production exceeded $CH_4$ oxidation (net $CH_4$ production).

**2.4 Resin bags**

Soil mineral nitrogen ($NO_3^-$ and $NH_4^+$) and phosphate ($PO_4^{3-}$) availability was assessed over four periods of 60 to 120 days using ion exchange resin bags. The bags were prepared by cutting nylon stockings into 10 cm-long pieces. One end was closed with a zip tie, and the bags were filled with 15 ml of mixed ion exchange resin beads (AmberLite MB20, Sigma-Aldrich; Tokyo, Japan). After closing the other end with a zip tie, the bags formed flat cylinders of approximately 4 cm in diameter. Before deployment, the resin bags were acid-washed in 10% HCl solution for 1 hour and rinsed multiple times with deionized water until the rinse water reached the same pH as the deionized water.

Resin bags were buried in the mineral soil at a depth of 5 cm below the litter layer in each of the four blocks and
four fertilizer treatments. Three bags were installed in each of the 16 individual plots on four occasions: February–May 2023; May–August 2023 immediately following the first fertilizer application in T2, T3 and T4; August–October 2023; and October 2023–February 2024 following the second fertilizer application in T3 and T4. Each new bag was placed at 90° angle from the previous position along the perimeter of a virtual circle with a radius of 20 cm.

After retrieval, the resin bags were rinsed in deionized water and stored either in a refrigerator in the laboratory or
in a cooler box during transport prior to extraction. The resin bags were extracted three times with 25 mL of 2 M NaCl, shaking for 1 hour each time. Extracts were analysed for $NO_3^-$ and $NH_4^+$ using flow injection analysis (Flow Injection Analyzer FI-5000V, Aqua Lab, Japan) and for $PO_4^{3-}$ colorimetrically. After extraction, the resin beads were removed from the bags, dried at 70 °C and weighed.

## 2.5 Dissolved organic carbon and total dissolved nitrogen in soil solutions

Six lysimetric pits (three in treatment T1 and three in T3 , distributed across three blocks) were installed in 2017 as part of another project to collect soil solutions. Solutions were collected using ceramic cup lysimeters connected to a vacuum pump set to -60 kPa of suction.

On February 21$^{st}$, August 17$^{th}$ and October 8$^{th}$, 2023, soil solutions were retrieved from two ceramic cup lysimeters installed at a depth of 15 cm in each pit. The collected solutions were stored at 4 °C and subsequently analysed for total
dissolved nitrogen (TDN) and dissolved organic carbon (DOC) using a total organic carbon analyser (TOC-L with TNM-L unit, Shimadzu, Japan).

## 2.6 Soil and climate ancillary data

Topsoil cores were collected in March and October 2023 using $5 \times 5$ cm sampling cylinders. In March, four samples per plot (16 per treatment) were taken, while in October, one sample per plot was collected. Before sampling, SWC was measured at
two positions 10 cm away from the sampling location to verify the calibration of the SWC probe. The fresh weight of the soil samples was recorded, after which they were air-dried, reweighed and sieved through a 2-mm mesh. Bulk density (BD)

was calculated as the ratio of oven-dried soil mass (measured on a subsample dried at 105 °C) to the volume of the sampling cylinders. SWC and bulk density (BD) were used to calculate air-filled porosity (AFP), assuming a particle density of 2.65 g cm$^{-3}$.

Soil pH (1:2.5 soil to water ratio) was measured on three soil samples (0–10 cm depth) in each plot (12 per treatment) after shaking the soil suspensions for 1 hour. Total soil carbon (C) and nitrogen (N) concentrations were determined on two soil samples (0–10 cm depth) from each plot (8 per treatment) using an elemental analyser (EA-Isolink CN, Thermo Fisher Scientific).

        Litter falls was collected biweekly from January 2023 to April 2023 (covering the leaf fall period) using two 50 ×

50 cm litter traps installed in each plot (8 per treatment). The collected litter was oven-dried at 65 °C and weighed. Composite samples for each treatment in each block were ground, and total C and N concentrations were measured as described for the soil samples.

        Air temperature ($T_A$; HMP155, Vaisala; Vantaa, Finland) and incident precipitation ($P_I$; tipping bucket rain gauge, ARG100/EC, Environmental Measurements Limited; North Shields, United Kingdom) were recorded every 10 seconds and

stored as 30 minutes averages for $T_A$ and cumulative sums for $P_I$. Measurements were taken using a datalogger (CR200X, Campbell Scientific, Logan, UT, USA) at a nearby weather station located 500 m from the plantation in an open area.

**2.7 Statistical analyses**

All data analyses were performed using R version 4.3.2 (R Core Team, 2023). Linear mixed-effects models (LMMs) were used to test the effects of fertilization and measurement date (fixed effects) on $F_{S-CH4}$, $T_{SOIL}$, AFP and SWC, with collar

identifiers included as a random effect. Similarly, LMMs were applied to soil $CH_4$ molar fraction, resin bag data and lysimeter data, using the location identifier as a random effect. For soil characteristics (BD, pH, total C and N), which were measured only once, and for $F_{T-CH4}$, which was not always measured on the same trees, block was included as a random effect. LMM were fitted using the 'lmerTest' package (Bates et al., 2015; Kuznetsova et al., 2017).. For litterfall and litter N content, simple linear models were used because all samples from each plot were combined, resulting in only one sample per

treatment per block. When residuals did not meet the assumption of normality, the dependent variables were rank-

transformed in the final models (Conover and Iman, 1981) using aligned rank transformation for nonparametric factorial analyses, as implemented in the 'ARTool' package (Wobbrock et al., 2011). Post-hoc contrasts were applied to test differences between treatments. The conclusions obtained from the rank-transformed data were consistent with those obtained from the raw data.

An LMM was also fitted to analyse the relationship between $F_{S-CH4}$ and AFP. Marginal ($R^2_m$) and conditional ($R^2_c$) coefficients of determination (Nakagawa and Schielzeth, 2013) were calculated using 'MUMIn' package (Bartoń, 2023).

    For each collar in each treatment, the number of measurement days with positive $CH_4$ fluxes was recorded. This number could range between 0 (all measured fluxes were negative for this collar) to 37 (all measured fluxes were positive for this collar). For the 24 collars in each treatment, both the median and the maximum of the number of days with positive flux

were calculated.

## 3 Results

### 3.1 Edaphic factors

Fertilization did not significantly affect soil bulk density, total carbon, or nitrogen concentrations ($p = 0.11$, 0.55, and 0.81, respectively) but acidified the soil, particularly in T3 and T4 (Table 1, $p < 0.001$). While the amount of litterfall did not differ

markedly between treatments ($0.73 \pm 0.03$ kg m$^{-2}$ on average), the nitrogen content of the litter was 9% higher in T3 and T4 litters compared to T1 and T2 (Table 1, $p < 0.001$).

**Table 1. Soil and litter characteristics under four fertilization treatments. Differences in bulk density (BD), pH, and total carbon and nitrogen concentrations in the top 10 cm of soil between the four fertilization treatments. Values are averaged by treatment and presented with standard error. The p-values from ANOVA applied to linear mixed-effects models (soil) or linear model (litter) on rank-transformed data are shown, along with n, the number of independent replicates in each treatment. Significant differences between fertilization levels ($p < 0.05$) are indicated by different lowercase letters.**

| Treatment N/P/K (kg ha$^{-1}$ yr$^{-1}$) | Soil | | | | Litter | |
|---|---|---|---|---|---|---|
| | BD (kg dm$^{-3}$) | pH | C (g kg$^{-1}$) | N (g kg$^{-1}$) | Amount (kg m$^{-2}$) | N (g kg$^{-1}$) |
| T1 | 1.44 | 5.90 | 3.3 | 0.37 | 0.66 | 15.6 |
| (none) | ± 0.02 | ± 0.07 c | ± 0.3 | ± 0.03 | ± 0.09 | ± 0.2 a |
| T2 | 1.48 | 5.41 | 3.8 | 0.39 | 0.77 | 16.1 |
| (37/22/50) | ± 0.02 | ± 0.10 b | ± 0.3 | ± 0.02 | ± 0.03 | ± 0.3 a |
| T3 | 1.49 | 4.89 | 3.7 | 0.39 | 0.77 | 17.4 |
| (90/40/85) | ± 0.01 | ± 0.07 a | ± 0.4 | ± 0.04 | ± 0.05 | ± 0.2 b |
| T4 | 1.47 | 4.95 | 3.9 | 0.39 | 0.71 | 17.3 |
| (153/68/144) | ± 0.03 | ± 0.07 a | ± 0.5 | ± 0.03 | ± 0.04 | ± 0.1 b |
| p-value and [n] | p = 0.11 [n = 20] | p < 0.001 [n = 12] | p = 0.55 [n = 8] | p = 0.81 [n = 8] | p = 0.66 [n = 4] | p < 0.001 [n = 4] |

Fertilization did not significantly affect SWC or AFP (Table A1, $p > 0.1$). The soil in T1 exhibit slightly but significantly higher temperatures at 10 cm depth compared to the other treatments (+0.5, +0.7, and +0.8 °C above T2, T3, and T4, respectively; Table A1, $p < 0.001$).

## 3.2 Soil methane flux

Seasonal rainfall influenced the AFP and soil CH$_4$ fluxes ($F_{S-CH4}$), with higher AFP and increased CH$_4$ uptake (more negative values) during the dry season compared to the rainy season (Fig. 1A–C). Lower CH$_4$ uptake was observed at 2.0 m from the planting rows compared to 0.7 m and 3.3 m in the fertilized treatments, likely reflecting spatial heterogeneity in fertilizer application, as fertilizer was broadcast by workers walking approximately 2 m from the planting rows. Significant differences in $F_{S-CH4}$ were observed across all dates and fertilization treatments, with fertilization decreasing soil CH$_4$ uptake

and increasing emissions (Fig. 1D, Table A1, p < 0.001). In heavily fertilized plots, the soil even transitioned from a net $CH_4$

sink to a net source during the rainy season. Out of 24 collars, six in T1 never showed positive $F_{S-CH4}$. The median number of

measurement days with positive $F_{S-CH4}$ was 2 (maximum of 7 days). For T2, T3, and T4, these numbers were three collars

(median: 3 days; maximum: 23 days), two collars (median: 7 days; maximum: 27 days), and one collar (median: 15 days;

maximum: 31 days), respectively. Some collars exhibited transient positive $F_{S-CH4}$, occasionally during the dry season,

without synchronization within the same treatment (Fig. A1).

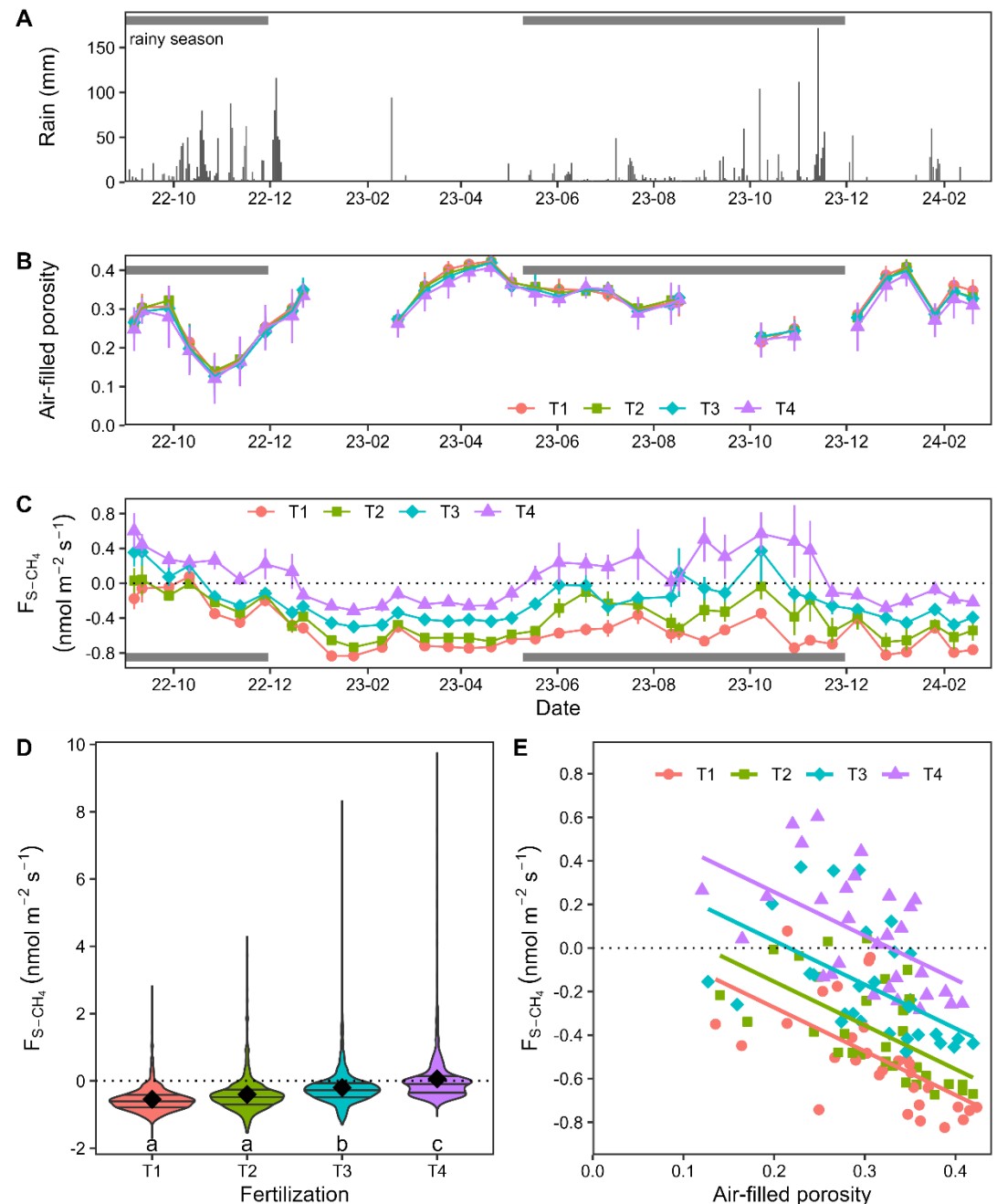


**Figure 1. Soil CH₄ fluxes over 1.5 years in an experimental rubber plantation with four levels of fertilization. (A) Daily rainfall; (B) soil air-filled porosity averaged by treatment with SEM bars (n = 24); (C) soil CH₄ fluxes (F$_{S-CH4}$) averaged by treatment with SEM bars (n=24); (D) violin plots showing the variability in F$_{S-CH4}$ within and between the four fertilization treatments (n = 888); and (E) linear regressions between air-filled porosity and F$_{S-CH4}$ averaged for each day across fertilization levels (n=37, statistics and**

**model parameters are provided in Table A2). Fertilization treatments are ranked from T1 (no fertilization) to T4 (highest fertilization level) and are represented by different colours in panels B–E and different symbols in panels B, C and E. Significant differences between fertilization treatments (p < 0.05) are indicated by different lowercase letters in panel D**

The average $CH_4$ uptake was higher in the non-fertilized treatment (T1: -0.54 ± 0.01 nmol $m^{-2}$ $s^{-1}$, n = 888, mean ± SEM) compared to T2 (-0.40 ± 0.02 nmol $m^{-2}$ $s^{-1}$) and T3 (-0.02 ± 0.12 nmol $m^{-2}$ $s^{-1}$). In T4, the average $F_{S-CH4}$ was positive (0.06 ± 0.03 nmol $m^{-2}$ $s^{-1}$). Across 37 measurement days from September 2022 to February 2024, the spatially averaged $F_{S-CH4}$ was positive on only one day for T1, two days for T2, six days for T3 and 20 days for T4. Fertilization increased the intercept of the relationships between $F_{S-CH4}$ and AFP while the slope remained consistent across treatments (Fig. 1E and Table A2).

### 3.3 Soil methane mole fraction

Soil $CH_4$ mole fractions ($[CH_4]_S$) between 0 and 10 cm depth decreased by an average of -35 ppb $cm^{-1}$ ($\Delta CH_4$) compared to ambient air (mole fraction of 1975 ppb on average) across all dates and locations in T1. An exception was observed in one pipe in August 2023 where $[CH_4]_S$ at 10 cm depth was higher than ambient air (2025 ppb, Fig. 2A). A similar trend was noted in T2, with a lesser decrease ($\Delta CH_4$ = -20 ppb $cm^{-1}$) and three occurrences of mole fractions above ambient air, including a hot spot of $CH_4$ accumulation during the rainy season (October 2023, lowest AFP, 3759 ppb). In T3 and T4, $[CH_4]_S$ increased between 0 and 10 cm with $\Delta CH_4$ values of 14 and 7 ppb $cm^{-1}$ on average, respectively (p = 0.04, Table A3), and hotspots of $CH_4$ accumulation (> 2500 ppb) occurring in October 2023 for both treatments. Overall, net $CH_4$ consumption dominated in T1 and T2 soils at a depth of 0 to 10 cm while net $CH_4$ production dominated in T3 and T4 soils.

At a depth of 10 to 40 cm, $[CH_4]_S$ decreased with $\Delta CH_4$ values of -18 ppb $cm^{-1}$ on average, with no significant differences between fertilization treatments (Fig. 2B and Table A3, p > 0.6). $[CH_4]_S$ at 40 cm depth was higher in October 2023 during the wet season than at the two other dates.

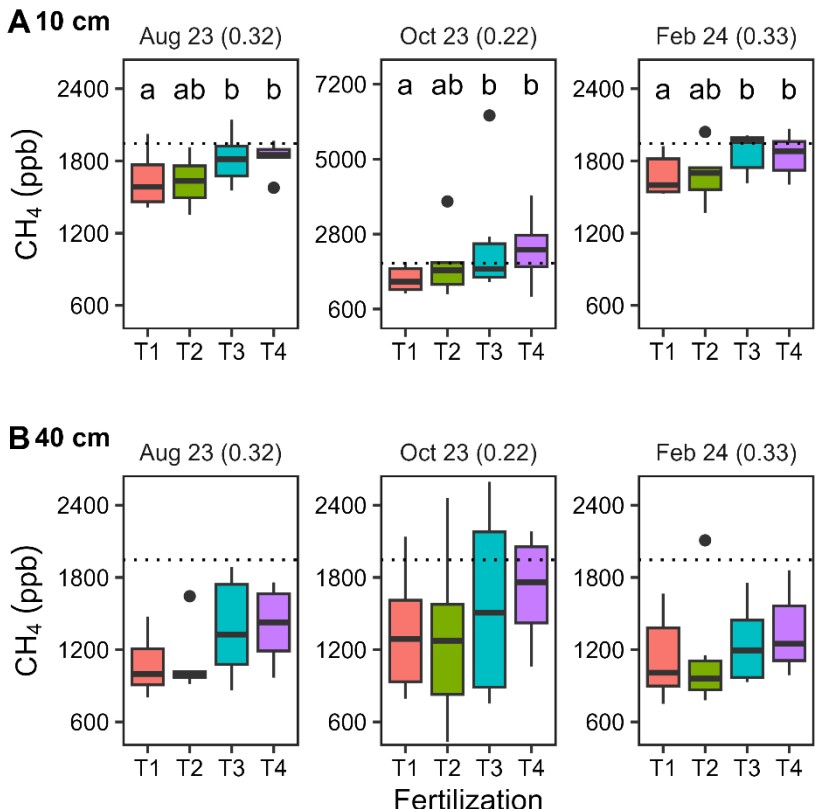

**Figure 2. Soil CH₄ mole fractions measured at two depths in different seasons. Boxplots (n = 6) illustrating soil CH₄ mole fractions at (A) 10 cm depth and (B) 140 cm depth on three different dates. Fertilization treatments are ranked from T1 (no fertilization) to T4 (highest fertilization level) and are shown in different colours. The boxes delimitate the interquartile range, the solid lines indicate the median and the points represent outliers. Note: the scale is different for the topsoil (10 cm) in October 2023.**

### 3.4 Inorganic nitrogen and phosphorus dynamics

Ammonium ($NH_4^+$), nitrate ($NO_3^-$) and phosphate ($PO_4^{3-}$) accumulated in resin bags, particularly those buried immediately after fertilization applications in May (T2, T3 and T4) and in October (T3 and T4 only; Fig. 3 and Table A4). Concentrations increased with fertilization levels ($p < 0.001$) with higher values in T3 and T4 compared to T1 and T2. Differences were less pronounced for $NH_4^+$ than for $NO_3^-$ and $PO_4^{3-}$. After May fertilization, resin bags collected 1.7 times more $NH_4^+$ in T2, 9.0 times more in T3 and 9.6 times more in T4 than in T1. Similarly, $NO_3^-$ concentrations increased by 13, 494 and 600 times, and $PO_4^{3-}$ concentrations by 16, 25 and 43 times, respectively. After the October fertilization, $NH_4^+$ concentrations increased

by 2.3 times in T2 (despite no fertilization), 7.4 times in T3 and 9.6 times in T4 compared to T1. $NO_3^-$ and $PO_4^{3-}$ concentrations also increased substantially. The lowest concentrations were recorded during the dry season (February to

May), and lower concentrations were observed in bags buried three months after fertilization (August) compared to those buried immediately after (May and October).

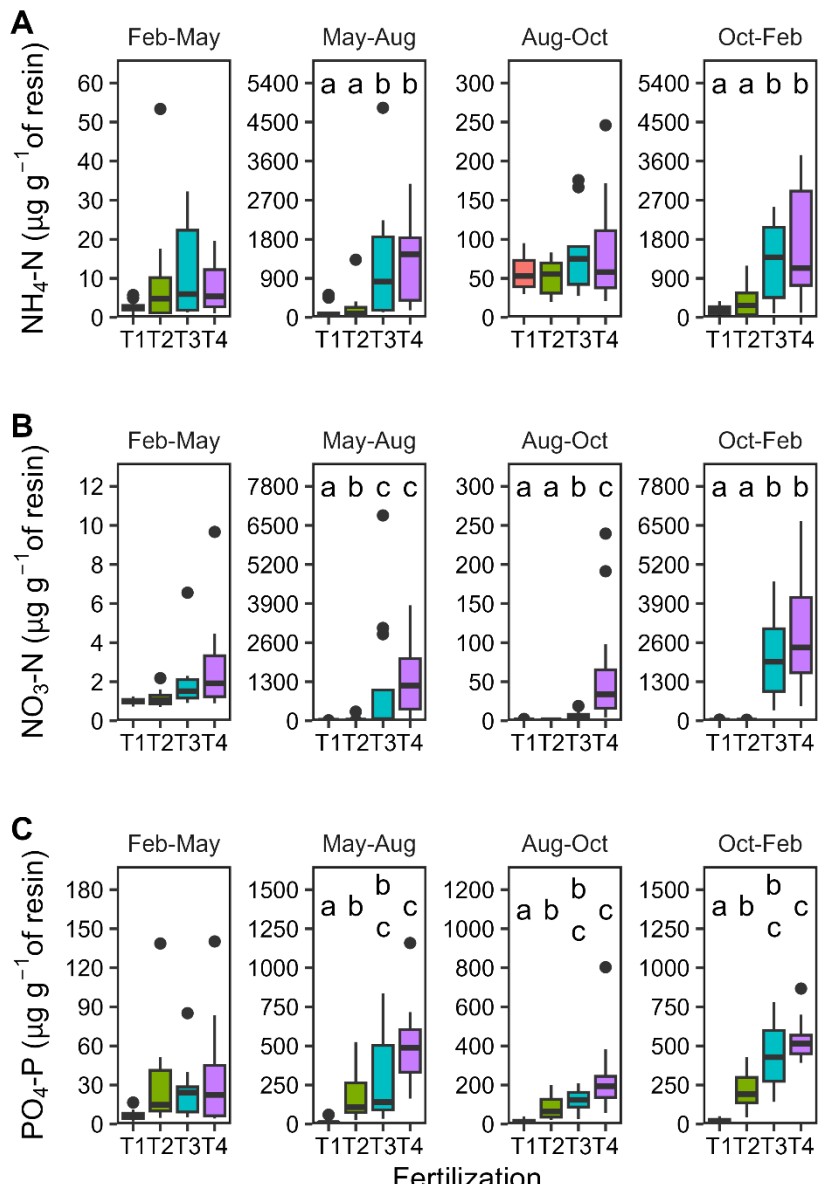

**Figure 3. Seasonal variations in mineral nitrogen and phosphate concentrations in resin bags. Boxplots (n = 12) illustrating concentrations in (A) NH4-N, (B) NO3-N and (C) PO4-P collected during four different seasons using resin bags buried at 5 cm soil depth. Fertilization treatments are ranked from T1 (no fertilization) to T4 (highest fertilization level) and are shown in different colours. The boxes delimitate the interquartile range, the solid lines indicate the median and the points represent outliers. Note: the scales differ between seasons to accommodate the large variations associated with rainfall seasonality (dry season from early December to early May) and fertilizer applications (in May for all treatments except T1, and additionally in October for T3 and T4).**

### 3.5 Total Dissolved nitrogen and dissolved organic carbon in lysimeter water

Total dissolved nitrogen (TDN) and dissolved organic carbon (DOC) in lysimeter waters collected at 15 cm depth (T1 and T3 only) showed no pronounced seasonal variations (p=0.08) but significant differences between treatments (p < 0.001, Fig. 4 and Table A5). On average, TDN and DOC concentrations were positively correlated (Spearman's rank correlation coefficient, $\rho$ = 0.61, p < 0.001, df = 33), with concentrations 3.6 and 4.5 times higher, respectively, in T3 compared to T1.

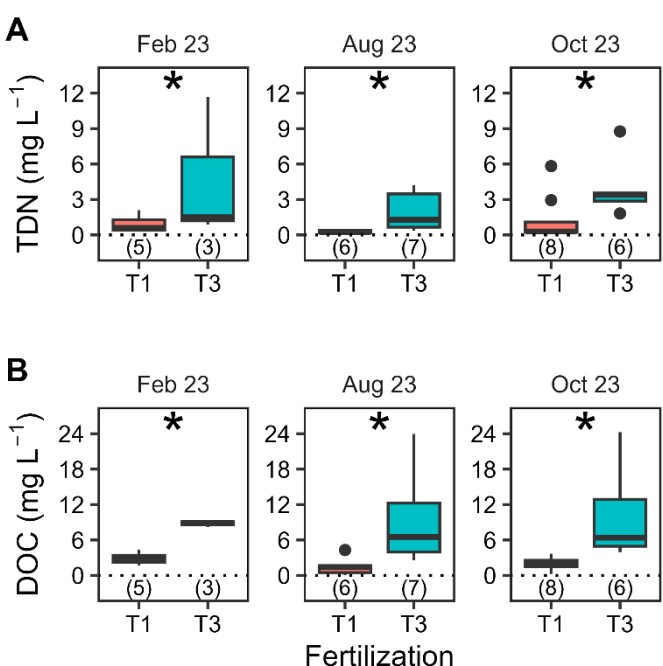

**Figure 4. Concentrations of total dissolved nitrogen and dissolved organic carbon in lysimeter waters. Boxplots illustrating concentrations of (A) total dissolved nitrogen (TDN) and (B) dissolved organic carbon (DOC) in lysimeter waters collected at 15 cm soil depth on three different dates. Fertilization treatments are ranked from T1 (no fertilization) and T3 and are shown in different colours. The boxes delimitate the interquartile range, the solid lines indicate the median and the points represent outliers.**

**The number of samples analysed varied and is indicated in parentheses below each box. Note: a minimum of five values is formally required to produce a boxplot, which was not the case for T3 in February 2023.**

### 3.6 Trunk methane flux.

Most rubber tree trunks emitted $CH_4$ (positive $F_{T-CH4}$), although a few oxidized it (negative $F_{T-CH4}$). $F_{T-CH4}$ ranged from -0.04 to 0.93 nmol $m^{-2}$ $s^{-1}$, with a median of 0.05 (n = 233). For most trees, $F_{T-CH4}$ was highest near the base (40–60 cm from the ground) and decreased slightly with height along the trunk, although the differences between heights were not significant (p=0.34).

Upscaled trunk $CH_4$ fluxes ranged from -0.10 to 0.51 nmol $s^{-1}$ per tree, with no significant fertilization effect but marked differences between measurement dates (Fig. 5 and Table A6, p = 0.91 and < 0.001 respectively). The upscaled fluxes increased substantially between August 2023 (0.044 ± 0.008 nmol $s^{-1}$ $tree^{-1}$, mean with SEM, n = 45) and October 2023 during the wet season (0.10 ± 0.01, n = 33), and again between October 2023 and February 2024 during the dry season (0.19 ± 0.01, n = 33). Out of 45 trunks, seven were net $CH_4$ oxidizers in August 2023, but all trunks were net $CH_4$ emitters in subsequent measurements.

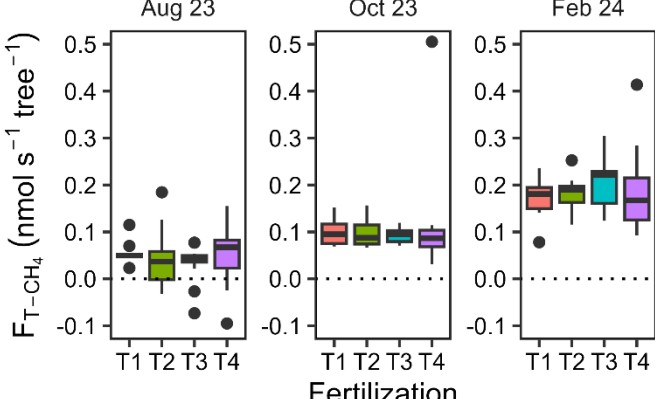

**Figure 5. Trunk $CH_4$ fluxes scaled to tree levels across seasons. Boxplots illustrating trunk $CH_4$ fluxes scaled to tree level, measured on 8 to 13 trees per treatment and on three different dates in August 2023, October 2023 and February 2024. Fertilization treatments are ranked from T1 (no fertilization) to T4 (highest fertilization level) and are shown in different colours. The boxes delimitate the interquartile range, the solid lines indicate the median and the points represent outliers**

When $F_{T-CH4}$ was scaled by tree density to allow comparison with $F_{S-CH4}$, both expressed on a soil surface basis, $F_{T-CH4}$ offset soil $CH_4$ uptake by less than 0.5% in T1 and T2, and 1.8% in T3 in August 2023. In T4, however, trunk emissions accounted

for 3.6% of the combined net $CH_4$ emissions from trunks and soil. In October 2023, $F_{T-CH4}$ offset soil $CH_4$ uptake by 1.5% in T1 and 14% in T2 and contributed 1.6% in T3 and 0.9% in T4 to the combined net $CH_4$ emissions. In February 2024,the proportion of soil $CH_4$ uptake offset by $F_{T-CH4}$ was 1.2%, 1.7%, 2.4%, and 4.4% for T1, T2, T3, and T4 respectively.

### 3.7 Annual soil methane uptake

Annual soil $CH_4$ fluxes, upscaled to the plantation level for the periods from September 6, 2022 to September 5, 2023 and

from February 20, 2023 to February 19, 2024, decreased significantly with increasing levels of fertilization ($p < 0.001$). The differences between T1 (no fertilizer application) and T2 were moderate (Table 2). However, the soil $CH_4$ sink in T3 was reduced by approximately 60%, and heavy fertilizer application in T4 even transformed the soil from a net $CH_4$ sink into a net $CH_4$ source.

**Table 2. Net annual soil $CH_4$ uptake. Cumulative annual soil $CH_4$ fluxes measured from September 6, 2022 to September 5, 2023,**
**and from February 20, 2023 and February 19, 2024, averaged by treatment (n = 4 blocks) with standard errors. The p-values from ANOVA applied to linear mixed-effects models on rank-transformed soil fluxes are shown, along with n, the number of independent replicates per treatment. Significant differences between fertilization levels (p < 0.05) are indicated by different lowercase letters.**

| Treatment (N/P/K) (kg ha$^{-1}$ yr$^{-1}$) | Soil $CH_4$ flux (kg $CH_4$ ha$^{-1}$ yr$^{-1}$) | |
|---|---|---|
| | Sep 6, 2022 to Sep 5, 2023 | Feb 20, 2023 to Feb 19, 2024 |
| T1 (none) | - 2.6 ± 0.2 a | - 3.1 ± 0.2 a |
| T2 (37/22/50) | - 2.0 ± 0.2 a | - 2.2 ± 0.4 b |
| T3 (90/40/85) | - 1.1 ± 0.2 b | - 1.1 ± 0.3 c |
| T4 (153/68/144) | + 0.2 ± 0.3 c | + 0.3 ± 0.5 d |
| p-value and [n] | p < 0.001 [n = 24] | p < 0.001 [n = 24] |

## 4 Discussion

### 4.1 Fertilizer application reduces soil methane uptake

Soil $CH_4$ fluxes measured in this study were within the range previously reported for mature rubber plantations in Southeast Asia (Sumatra, Yunnan). These studies reported daily averages ranging from -0.71 to 1.13 nmol m$^{-2}$ s$^{-1}$ and annual fluxes between -3.1 and -0.2 kg $CH_4$ ha$^{-1}$ yr$^{-1}$ (Aini et al., 2020; Ishizuka et al., 2002, 2005; Lang et al., 2017, 2019; Werner et al., 2006; Zhou et al., 2021). These broad ranges likely reflect differences in edaphic factors across sites—such as soil texture, porosity, and infiltrability—which influence gas diffusion and soil moisture, and thereby affect $CH_4$ consumption and production. However, based on our results, variability in fertilizer application could also explained some of these differences, as not all plantations in earlier studies were fertilized.

Seasonal variation in $F_{S-CH4}$ were closely linked to changes in AFP. Low AFP restricts gas diffusion from the atmosphere into the soil, limiting the availability of $O_2$ and $CH_4$ for methanotrophic bacteria (Hu et al., 2023; Kruse et al., 1996). In our study, differences in $F_{S-CH4}$ between fertilization treatments were not driven by differences in AFP, which might have resulted from differences in tree water use (Qiu et al., 2024). Instead, they were clearly related to the rate of fertilizer applications.

The vertical profile of $[CH_4]_S$ indicated that $CH_4$ oxidation occurred throughout the soil profile, at least down to a depth of 40 cm. Negative concentration gradients between 10 and 40 cm depth were observed across all seasons and treatments. Contrary to previous suggestions (Wang and Ineson, 2003), high concentrations of inorganic nitrogen in the topsoil of fertilized plots did not displace $CH_4$ oxidation to deeper layers. Instead, fertilizer application inhibited $CH_4$ oxidation in the topsoil when AFP was high. This was evident from the lower $[CH_4]_S$ measured at 10 cm depth in T1 and T2 compared to T3 and T4 in August 2023 and February 2024.

### 4.2 Effects of fertilizer on methane oxidation

Previous studies examining the effects of nitrogen fertilizer applications on $CH_4$ consumption in forest soils have reported contradictory results. Some studies reported reduced $CH_4$ oxidation in response to nitrogen addition (Castro et al., 1994;

Chan et al., 2005; Jassal et al., 2011; Steudler et al., 1989; Zhang et al., 2008), while others reported stimulation, suggesting that nitrogen-limited methanotrophic activity could benefit from fertilizer application (Börjesson and Nohrstedt, 2000; Hassler et al., 2015; Martinson et al., 2021; Papen et al., 2001; Qiu et al., 2024; Veldkamp et al., 2013). Therefore, nitrogen can potentially inhibit or stimulate $CH_4$ consumption in soils (Bodelier and Laanbroek, 2004). In our experimental plantation, *ex-situ* soil incubations showed that fertilization suppressed soil $CH_4$ oxidation potentials throughout the soil profile, at least down to 60 cm depth (Murase et al., 2024). We did not observe the biphasic dose-response relationship often reported, where low nitrogen inputs stimulate soil $CH_4$ uptake and higher doses inhibit it (Aronson and Helliker, 2010; Cen et al., 2024). In our study, even low fertilizer application rates (T2) failed to stimulate $CH_4$ uptake, suggesting that the soil may already have been nitrogen-saturated. The long term application of fertilizers (8 years at our site) may have increased the sensitivity of the soil methanotrophic communities to nitrogen addition (Aronson and Helliker, 2010). The response factor of soil $CH_4$ flux to nitrogen input, calculated as the ratio of the difference in $CH_4$ flux between each fertilized treatment and T1 to the annual nitrogen input, was 0.02 kg $CH_4$ kg $N^{-1}$ across all treatments. This value aligns with those reported for nitrogen-saturated forest soils under high nitrogen inputs (Cen et al., 2024), and is consistent with the previous land use (coconut plantation) and the applied fertilization levels. Furthermore, increased nitrogen input via litter decomposition may contribute to reduced $CH_4$ uptake, as has been suggested for tropical forest soils (Gao et al., 2022).

While the accumulation of $NH_4^+$ in resin bags, particularly in T3 and T4 after fertilization, suggests it may contribute to inhibition, the consistent accumulation of $NO_3^-$ at all deployment dates indicates that $NO_3^-$ toxicity could also play a significant role (Mochizuki et al., 2012; Reay and Nedwell, 2004; Wang and Ineson, 2003). This effect can be exacerbated if $NO_3^-$ is reduced to $NO_2^-$ in anaerobic microsites. Additionally, the decrease in soil pH observed from T1 to T4 with increased nitrogen addition is therefore another factor known to inhibit soil $CH_4$ oxidation (Benstead and King, 2001). Although methanotrophs can occur in both acidic and alkaline habitats, they usually grow better at neutral pH (Chowdhury and Dick, 2013; Hanson and Hanson, 1996; Whittenbury et al., 1970; Yao et al., 2023). Liming agricultural soils to raise their pH is known to stimulate soil $CH_4$ oxidation (Abalos et al., 2020; Fonseca de Souza et al., 2025). Large concentrations of $PO_4^{3-}$ also accumulated in resin bags across all dates and fertilized treatments. However, the mechanisms underlying the

interaction between phosphorus and nitrogen and its effects on $CH_4$ oxidation remained poorly understood (Veraart et al., 2015; Zheng et al., 2016).

### 4.3 Fertilizer application increase soil methane production

Our study revealed that $CH_4$ production occurred in the soil, particularly during the wet season, as indicated by positive $F_{S-CH4}$ values and soil $CH_4$ mole fraction ($[CH_4]_S$) exceeding those of ambient air. Therefore, $F_{S-CH4}$ reflected the net balance between $CH_4$ production and $CH_4$ consumption. More frequent and intense soil $CH_4$ emissions, along with higher soil $CH_4$ mole fractions in the fertilized treatments— particularly in T3 and T4—suggest that fertilizer application not only suppressed methanotrophic activity but also stimulated methanogenesis, as recently observed in an irrigated and fertilized sapling

plantation on a lowland soil (Rabbai et al., 2024), despite concurrent soil acidification. Like methanotrophs, methanogens typically grow better at neutral pH, and methanogenesis has been showed to be limited under low pH conditions in anoxic sediments (Garcia et al., 2000; Phelps and Zeikus, 1984). Our findings contrast with previous studies that reported decreased methanogenesis following either $NO_3^-$ addition to rice paddy soils and wetland sediments (Kim et al., 2015; Roy and Conrad, 1999) or lowering soil pH of peatland and rice paddy soils (Wang et al., 1993; Ye et al., 2012). However, our results are

consistent with a recent finding showing that combined nitrogen and phosphorus amendments increased $CH_4$ production in incubated soils from boreal peatland (Byun et al., 2025).

Banger et al. (2012) suggested that nitrogen fertilizers may stimulate $CH_4$ production both by alleviating nitrogen limitation to methanogens and by increasing crop growth, thereby enhancing the availability of carbon substrates for methanogenesis. In addition to anaerobic conditions, methanogenesis actually requires organic substrates derived from root

exudates, buried litter fragments or litter leachates—all products of plant photosynthesis (Bertora et al., 2018; Lu and Conrad, 2005; Minoda et al., 1996; Minoda and Kimura, 1994; Whiting and Chanton, 1993). $CH_4$ production potential has been linked to DOC concentration in wetland soils (Liu et al., 2011). The higher DOC concentrations observed in lysimeter water at 15 cm depth in T3 compared to T1 align with higher $CH_4$ production in T3 compared to T1. The cause of the elevated DOC concentration remained unclear, but phosphorus has been shown to enhance fine root biomass in P-limited

tropical secondary forests and tree plantations, potentially increasing root exudation (Zheng et al., 2016; Zhu et al., 2013).

Furthermore, trees in T3 and T4 produced litter with higher nitrogen content, which likely decomposes more rapidly, especially if nitrogen addition stimulates microbial mineralization (Cornwell et al., 2008; Hobbie, 2005; Melillo et al., 1982).

Interestingly, net $CH_4$ production mainly occurred in the top soil layer in our study. When soil $CH_4$ concentrations exceeded ambient level, they were consistently higher at 10 cm than at 40 cm depth. Methanogenesis requires anaerobic conditions, typically found in water-saturated soils (Epron et al., 2016; Smith et al., 2003). However, except for brief periods following heavy rainfall, the soil was not flooded, and the AFP of the top soil layer remained above 0.1. This suggests the presence of anaerobic microsites in the topsoil, where $O_2$ consumption by root and microbial respiration outpaces the diffusive flux of $O_2$ from the atmosphere. Such microsites are commonly found in otherwise oxic soils (Lacroix et al., 2023; Sexstone et al., 1985; Smith et al., 2003; Teh et al., 2005). Using a isotope-based pool dilution technique, von Fischer and Hedin (2007) demonstrated that small diversions of organic carbon flow from non-methanogenic to methanogenic pathways, likely occurring in anaerobic microsites, can transform soil cores from a net $CH_4$ sink into a net $CH_4$ source. Higher methanogenic activity and greater abundance of Archaea was found in soil cores containing larger amounts of fresh organic matter compared to those with lower amounts when anaerobically incubated (Wachinger et al., 2000). The transient nature of positive $F_{S-CH4}$ values and the lack of synchronicity between collars likely reflect the dynamic nature of these microsites, which are driven by small-scale spatial and temporal variations in soil $O_2$ supply and demand (Lacroix et al., 2023). Variations of $O_2$ demand could arise from microbial respiration, potentially driven by soil invertebrates, such as leaf-cutting ants and earthworms, that bury plant debris or organic matter (Caiafa et al., 2023; Kammann et al., 2009; Mehring et al., 2021). Termite colonies or Scarabaeidae larvae might also contribute to localised hotspots of $CH_4$ production (Hackstein and Stumm, 1994; Räsänen et al., 2023; Rasmussen and Khalil, 1983). Although we did not investigate soil invertebrates in this study, termite mounds and ant nests were present in the plantation. Future research should explore the long-term impacts of fertilization on all soil microbial and invertebrate communities, not only methanotrophs and methanogens.

**4.4 The $CH_4$ emitted by the rubber tree trunks is probably of internal origin**

Rubber trees at our site emitted $CH_4$, which could either be transported from the soil or produced internally by methanogenic archaea (Barba et al., 2019b; Covey and Megonigal, 2019). Interestingly, while soil $CH_4$ emissions and elevated soil $CH_4$

mole fractions were primarily observed during the wet season, the highest emissions from tree trunks occurred in February, during the dry season. Additionally, trunk $CH_4$ emissions did not differ significantly between fertilization treatments, despite higher $CH_4$ production in the soils of heavily fertilized plots (T3 and T4). These findings suggest that $CH_4$ emitted by rubber trees, despite a slight decreasing trend with height along the trunk, may have been produced internally rather than transported from the soil.

Trunk $CH_4$ emissions are commonly observed in large trees and positively correlated with trunk diameter when $CH_4$ production occurred in the heartwood (Epron et al., 2023; Pitz et al., 2018; Wang et al., 2017). This is because the anoxic conditions required for methanogenesis are more likely to develop as the length of the $O_2$ diffusion path increases or when water begins to accumulate in the heartwood (wetwood). With tree ageing, the onset of heartwood decay can provide substrates for methanogens, further facilitating $CH_4$ production (Epron and Mochidome, 2024).

In our study area, rubber trees are tapped for latex collection annually from May to late February. Previous studies have shown that the respiration rate of inner bark tissue in rubber trees increases after tapping resumes and decreases during the resting period (Annamalainathan et al., 2001). Trunk $CH_4$ emissions were lowest in August (three months after tapping resumed), intermediate in October (five months after) and highest in February (nine months after). Although this temporal pattern could be coincidental, it is possible that the intense physiological activity associated with latex regeneration in the inner bark consumes substantial amounts of $O_2$, reducing the quantity available for diffusion into the trunk. This reduction in $O_2$ could create localized anoxic conditions, facilitating $CH_4$ production in the wood.

**4.5 Implications for the greenhouse gas budget of the Agriculture, Forestry and Other Land Use sector.**

Our study provides new insights into the dual effects of fertilization on $CH_4$ dynamics in rubber plantations, demonstrating that it can simultaneously reduce $CH_4$ uptake and increase $CH_4$ emission. We acknowledge the potential biases associated with interpolating biweekly manual soil flux measurements, particularly given the possibility of high short-term temporal variability. Automated measurements would have been valuable for capturing flux dynamics at finer temporal scales (Barba et al., 2019a; Gana et al., 2018). However, implementing such a system would have been challenging in our experimental plantation, which included four blocks and four fertilizer treatments spread over a 9-ha area, with large distances between

chambers and the gas analyzer. Despite these limitations, our findings provide indicative estimates that advance our

understanding of the complex interactions between land management practices and greenhouse gas fluxes in tropical

agricultural systems. In the Agriculture, Forestry and Other Land Use (AFOLU) sector, only positive $CH_4$ fluxes are

typically reported as greenhouse gas emission. Negative $CH_4$ emissions (atmospheric $CH_4$ removal) are not accounted for.

However, the loss of soil $CH_4$ oxidation potential caused by agricultural practice is equally important. Conversely, practices

that preserve or enhance soil $CH_4$ uptake could serve as effective mitigation strategies.

Given that T3 represents the recommended fertilizer application rate for mature rubber plantations in Thailand, as

advised by Thai public institutions, and that 40% of rubber farmers exceed this recommendation, as represented by T4

(Chambon et al., 2018), the net $CH_4$ uptake by soils of rubber plantations in Thailand is estimated at approximately -0.6 kg

$CH_4$ ha$^{-1}$ yr$^{-1}$ (based on Table 2). Reducing fertilization to the levels applied in T2 (rational fertilization) could increase the

net $CH_4$ sink by a factor of 3.5, reaching 2.1 kg $CH_4$ ha$^{-1}$ yr$^{-1}$. With rubber plantations covering 39,000 km$^2$ in Thailand in

2021 (IRSG, 2023), such a reduction in fertilizer application could enhance the net soil $CH_4$ sink by approximately 5.9 Gg

$CH_4$ yr$^{-1}$. This corresponds to more than 0.5 Tg $CO_2$-eq per year, given the high 20-year global warming potential (GWP) of

$CH_4$, which is more than 80 times that of $CO_2$ (IPCC, 2021). If all else is equal, the mitigation potential for the whole

Southeast Asia would be four times higher than that estimated for Thailand, since rubber plantations in Thailand represent

only 25% of the area under rubber cultivation in all of Southeast Asia. There are, however, limitations to this scaling-up

estimate. For instance, this study was conducted at a single site, and the response of soil $CH_4$ efflux to fertilizer application

may vary across the different physiographic regions of Thailand due to differences in climatic and edaphic conditions

(Rabbai et al., 2024). Specifically, the documented response for the sandy-textured soil at our site may differ from those for

soils with higher clay contents, which are expected to exhibit more reductive microsites, or from those of drained peatland.

However, to our knowledge, this experimental site is the only one in Thailand—and possibly in all of Southeast Asia—

530 actively testing different fertilization levels on mature rubber plantations. Therefore, the estimated potential of atmospheric

methane removal remains speculative and should be considered as a first approximation to encourage further research in this

direction.

Reducing fertilization in rubber plantation is thus an effective mechanism for atmospheric $CH_4$ removal. Our results nevertheless suggest that its potential to offset greenhouse gas emission from other agricultural activities in Southeast Asia, such as rice cultivation—the primary contributor to greenhouse gas emissions from the Agriculture sector, with 30 Tg $CO_2$eq yr$^{-1}$ in Thailand (Saiyasitpanich et al., 2024)— is limited. However, Tang et al. (Tang et al., 2024) have recently documented the stimulation of $CH_4$ emissions from rice fields by nitrogen fertilization at the global scale. Applying rational fertilization practices to other tree plantations and cropping systems worldwide could thus contribute to curb the increase in atmospheric $CH_4$ concentration. However, to convince policy makers, local authorities and producers that implementing rational fertilization practices is a credible pathway to enhance atmospheric $CH_4$ removal, it is essential to ensure that such practices do not compromise yields and stakeholder's incomes. This was the case for the rubber plantation at our site (Table A7) but remained to be confirmed for rubber plantations in other pedoclimatic context and for other agricultural land-uses.

**5 Conclusions**

The rapid expansion of rubber cultivation, driven by the demand for natural rubber in the tire industry constitutes a significant land-use change in Southeast Asia. Despite fertilization been a common practice in rubber plantations, its impact on soil methane ($CH_4$) dynamics remained poorly understood. Our study demonstrates that fertilization not only reduces soil $CH_4$ consumption but also increases $CH_4$ production, transforming rubber plantations from a net $CH_4$ sink into a source. Implementing rational fertilization practices could enhance atmospheric $CH_4$ removal. However, its overall impact on greenhouse gas emissions from the agricultural sector in Southeast Asia would remained modest, unless it is extended to other tree plantations and cropping systems. Moreover, to fully understand the impact of reduced fertilizer applications on greenhouse gas budgets, further research should also evaluate possible reductions in nitrous oxide ($N_2O$) emissions from soil, as $N_2O$ is another potent greenhouse gas. The scalability of mitigation strategies should also be assessed under varying climatic and management conditions.

**Appendix**

**Table A1.Soil CH$_4$ fluxes (Figure 1D). Summary of linear mixed models (LMMs) analysing the effects of fertilization, measurement dates, and their interactions (fixed effects) on rank-transformed soil CH$_4$ fluxes (F$_{S-CH4}$), soil temperature (T$_{SOIL}$), soil water content (SWC) and air-filled porosity (AFP). Collar identifier was included as random effects.**

| Response variable | Explanatory factors (fixed effects) | p-values |
|---|---|---|
| F$_{S-CH4}$ [n=3552] | Fertilization [df = 3] | $4.6 \times 10^{-14}$ |
| | Date [df = 36] | $< 2 \times 10^{-16}$ |
| | Fertilization $\times$ Date [df = 108] | $< 2 \times 10^{-16}$ |
| | | |
| T$_{SOIL}$ [n=3511] | Fertilization [df = 3] | $2.4 \times 10^{-5}$ |
| | Date [df = 36] | $< 2 \times 10^{-16}$ |
| | Fertilization $\times$ Date [df = 108] | $5.9 \times 10^{-12}$ |
| | | |
| SWC [n=2828] | Fertilization [df = 3] | 0.13 |
| | Date [df = 29] | $< 2 \times 10^{-16}$ |
| | Fertilization $\times$ Date [df = 108] | $3.8 \times 10^{-5}$ |
| AFP [n=2828] | Fertilization [df = 3] | 0.15 |
| | Date [df = 29] | $< 2 \times 10^{-16}$ |
| | Fertilization $\times$ Date [df = 108] | $3.8 \times 10^{-5}$ |

**Table A2. Relationships between soil CH$_4$ fluxes and air-filled porosity (Figure 1E). Summary of linear mixed models (LMMs) analysing the effect of air-filled porosity (AFP) on soil CH$_4$ fluxes (F$_{S-CH4}$), with fertilization treatment included as random intercept. Marginal (R$^2_m$) and conditional (R$^2_c$) coefficients of determination are reported in the final columns.**

| Explanatory variable | | Fixed effects | | Random effects | | Coefficients | |
|---|---|---|---|---|---|---|---|
| | | Estimate $\pm$ SE | p-values | Fertilization | Intercept | R$^2_m$ | R$^2_c$ |
| AFP | Intercept | $0.37 \pm 0.15$ | 0.041 | T1 | - 0.24 | 0.17 | 0.64 |
| [n=120] | Slope | $-2.01 \pm 0.27$ | $3.2 \times 10^{-11}$ | T2 | - 0.12 | | |
| | | | | T3 | + 0.07 | | |
| | | | | T4 | + 0.29 | | |

**Table A3. Soil CH$_4$ molar fractions and gradient in soil CH$_4$ molar fractions (Figure 2). Summary of linear mixed models (LMMs) analysing the effects of fertilization, measurement dates and their interactions on rank-transformed soil CH$_4$ molar fractions [CH$_4$] and gradient in soil CH$_4$ molar fractions ($\Delta$CH$_4$) at 10 and 40 cm depths. Collar identifier was included as random effects.**

| Response variable | Explanatory factors (fixed effects) | p-values |
|---|---|---|
| [CH$_4$]$_S$ at 10 cm | Fertilization | 0.05 |
| | Date | 0.18 |
| | Fertilization $\times$ Date | 0.04 |
| $\Delta$CH$_4$ 0-10 cm | Fertilization | 0.04 |
| | Date | 0.26 |
| | Fertilization $\times$ Date | 0.03 |
| [CH$_4$]$_S$ at 40 cm | Fertilization | 0.32 |
| | Date | 0.03 |
| | Fertilization $\times$ Date | 0.91 |
| $\Delta$CH$_4$ 10-40 cm | Fertilization | 0.64 |
| | Date | 0.60 |
| | Fertilization $\times$ Date | 0.18 |

**Table A4. Mineral nitrogen and phosphate concentrations in resin bags (Figure 3). Summary of linear mixed models (LMMs) analysing the effects of fertilization, measurement dates and their interactions (fixed effects) on rank-transformed $NH_4^+$, $NO_3^-$, and $PO_4^{3-}$ concentrations accumulated in resin bags. Location identifier was included as random effects.**

| Response variable | Explanatory factors (fixed effects) | p-values |
|---|---|---|
| $NH_4^+$ [n=192] | Fertilization [df = 3] | $5.5 \times 10^{-9}$ |
| | Date [df = 3] | $< 2 \times 10^{-16}$ |
| | Fertilization × Date [df = 9] | $2.6 \times 10^{-10}$ |
| | | |
| $NO_3^-$ [n=192] | Fertilization [df = 3] | $4.7 \times 10^{-12}$ |
| | Date [df = 3] | $< 2 \times 10^{-16}$ |
| | Fertilization × Date [df = 9] | $< 2 \times 10^{-16}$ |
| | | |
| $PO_4^{3-}$ [n=192] | Fertilization [df = 3] | $1.7 \times 10^{-15}$ |
| | Date [df = 3] | $< 2 \times 10^{-16}$ |
| | Fertilization × Date [df = 9] | $< 2 \times 10^{-16}$ |

**Table A5. Concentrations of total dissolved nitrogen and dissolved organic carbon in lysimeter waters (Figure 4). Summary of**
575 **linear mixed models (LMMs) analysing the effects of fertilization, measurement dates and their interactions (fixed effects) on rank-transformed total dissolved nitrogen (TDN) and dissolved organic carbon (DOC) concentrations in lysimeter water. Location identifier was included as random effect.**

| Response variable | Explanatory factors (fixed effects) | p-values |
|---|---|---|
| TDN [n=35] | Fertilization [df = 1] | $9.1 \times 10^{-4}$ |
| | Date [df = 2] | 0.18 |
| | Fertilization × Date [df = 2] | 0.41 |
| | | |
| DOC [n=35] | Fertilization [df = 1] | $4.2 \times 10^{-8}$ |
| | Date [df = 2] | 0.98 |
| | Fertilization × Date [df = 2] | 0.49 |
| | | |

**Table A6. Trunk CH$_4$ fluxes scaled to tree levels across seasons (Figure 5). Summary of linear mixed models (LMMs) analysing the effects of fertilization, measurement dates and their interaction (fixed effects) on rank-transformed trunk CH$_4$ fluxes (F$_{T-CH4}$). Block was included as random effect.**

| Response variable | Explanatory factors (fixed effects) | p-values |
|---|---|---|
| F$_{T-CH4}$ [n=111] | Fertilization [df = 3] | 0.91 |
| | Date [df = 2] | $< 2 \times 10^{-16}$ |
| | Fertilization × Date [df = 6] | 0.71 |

**Table A7. Effect of the fertilization on the cumulative dry latex yield over 10 years after the beginning of tapping (May 2014–February 2024) expressed in kg tree$^{-1}$. Values are averaged by treatment and presented with standard error. The p-value is from ANOVA applied to a linear model.**

| Treatments | Cumulative latex yield (kg tree$^{-1}$) 2014–2024 |
|---|---|
| T1 | 49 ± 3 |
| T2 | 55 ± 2 |
| T3 | 54 ± 1 |
| T4 | 55 ± 1 |
| p-value and [n] | 0.085 [n = 4] |

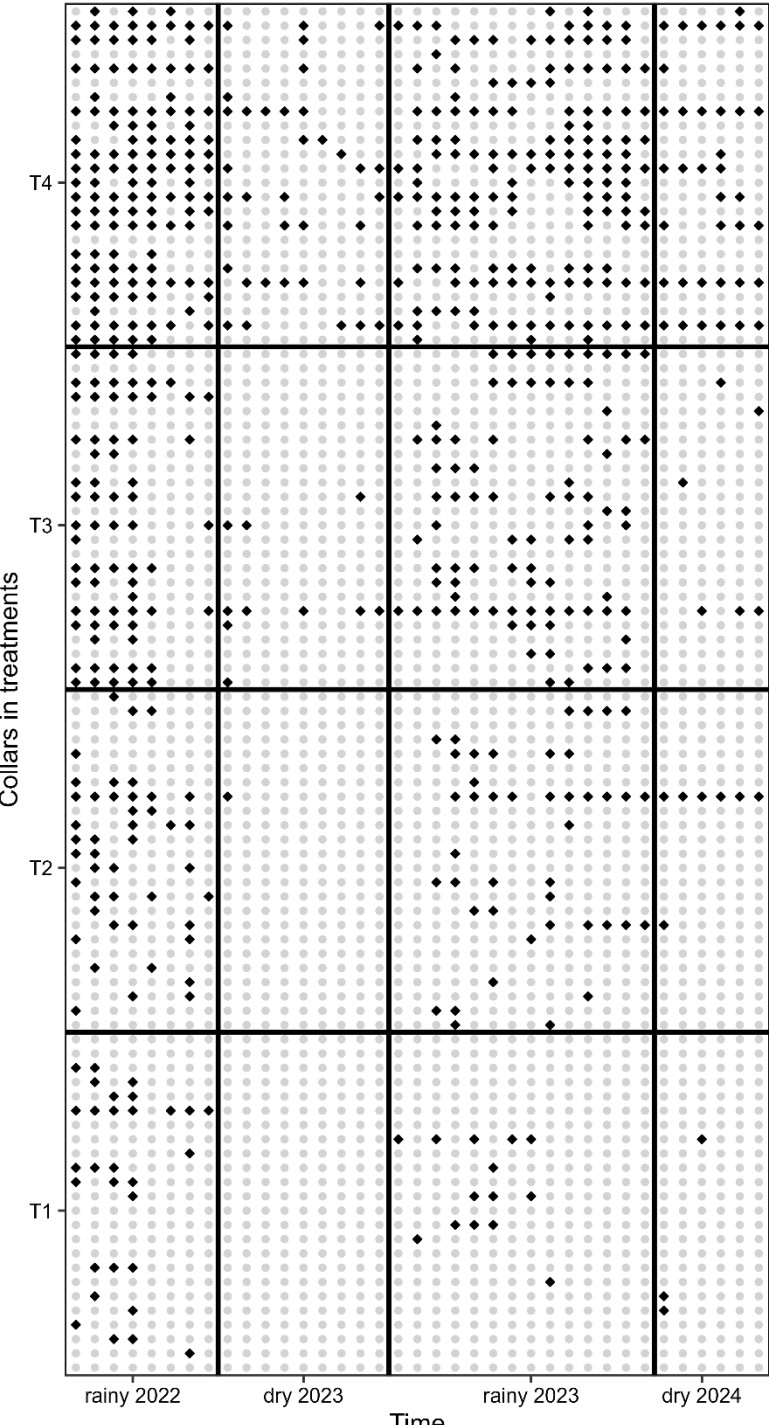

**Figure A1. Spatial and temporal change of the direction of soil methane fluxes. Green circles represent negative fluxes (net CH$_4$ uptake), while red diamonds indicate positive fluxes (net CH$_4$ emission). Spatial variation corresponds to the different collars (y-axis, 96 collars, 24 per treatment, ranked from T1 to T4), and temporal changes reflect measurements over time (x-axis, 37 dates).**

## Data availability

The data used in this study are available at the Dryad repository (DOI: 10.5061/dryad.59zw3r2jx)

## Author contribution

DE lead the research. DE, OD, YN, PK, and KS designed the research. DE, RC, ZW, OD, MS, SKP, TM, JS, WAA, and JM performed the research. DE analysed the data. DE wrote the manuscript that was critically revised by all co-authors.

## Competing interests

The setup of the experimental site was financially supported by Yara International, although this research project itself received no funding from Yara. The authors declare that they have no conflict of interest.

## Acknowledgments

The authors express their gratitude to the Faculty of Agriculture of Kasetsart University, Khampaeng Sean campus, for providing access to the Sithiporn Kridakara Research Station. Special thanks are extended to the staff of Sithiporn Kridakara Research Station and the DORAS research center for their invaluable contribution to the fieldwork, particularly Jeerapan Tipparat, Phetrada Kayankit, Natthaworn Kahohem, Jutamas Merasanud, Chalermchart Wongleecharoen, Rungtawan Thabkhum.

**Financial support**

This research was supported by the KAKENHI Grant-in-Aid for the Promotion of Joint International Research (Fostering Joint International Research B, grant no. 21KK0114). Additional supports were provided by the Office of the Ministry of Higher Education, Science, Research and Innovation; and the Thailand Science Research and Innovation through the Kasetsart University Reinventing University Program 2023.

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
