# Peer review of "Fertilization turns a rubber plantation from sink to methane source"

_EGUsphere, 2025_

## Author Response (AR1)

**Review 1**

This is an interesting study tackling an important emerging issue; namely, the potential impacts of rubber plantation expansion on biogeochemical cycling of key compounds, such as methane (CH4). The manuscript was generally well-written and the experimental design was sound. The strength of this paper is in its mechanistic focus, which will enable the biogeochemical and modelling community to develop better predictive frameworks for understanding how rubber production systems contribute to land-atmosphere interactions. In particular, the paper provides strong evidence for the impacts of varying N inputs on soil CH4 flux, characterises temporal/seasonal trends in CH4 flux, and explores the relationship between CH4 flux and key environmental variables (e.g. air-filled pore space).

Dear Professor Yit Arn The

Thank you for your positive and constructive feedback on our manuscript.

However, what I am less certain of is if the data collected here is sufficient to construct a whole ecosystem CH4 budget (see points 6 and 17). There are two reasons for this criticism: first, while soil CH4 flux is collected at regular intervals over 1.5 years, stem CH4 flux is only measured at 3 timepoints. This is not sufficient to infer CH4 emissions from tree stems for the entire observation period. Second, given that there are an increasing number of studies that use eddy covariance or automated chamber measurements to collect quasi-continuous CH4 measurements, it is my view that manual chamber measurements are insufficient at this point in time to construct an accurate ecosystem CH4 budget from. The main reason for this is that CH4 flux has been proved to be highly heterogeneous in time, meaning that low frequency, manual chamber measurements have been shown to cause large underestimates in CH4 flux. If this paper were published 25 years ago, where eddy covariance and automated chamber measurements were uncommon, then I think it would be reasonable to construct a soil CH4 budget from these data. However, I am concerned that in 2025 that a manual chamber paper may not contain accurate enough data to construct a credible budget from, given what we now know about the temporal heterogeneity of CH4 flux and the increasing use of quasi-continuous measurement systems. That said, I still believe that the authors can still discuss what these observations could mean for rubber plantation CH4 budgets. One way to do this is to calculate the budget as a post-hoc exercise and clearly flag in the paper that these calculations represent a "first approximation" or "preliminary" budget (see point 17). This means that the authors could still present their budget numbers but could also transparently acknowledge the limitations of the methods they have used. This is a subtle but important difference, because the authors are not claiming a high level of accuracy for a "preliminary" budget, but are providing indicative numbers in order to advance knowledge.

We understand your concern and fully agree that the methods we used have limitations and that we need to better acknowledge them in the discussion. We now avoid using the word "budget" when refereeing to our work. Please see the more detailed responses to points 6 and 17 below.

However, we would like to mention that eddy covariance methods are not applicable to an experimental plantation with four blocks and four fertilizer treatments (16 plots) due to the large and spatially variable footprint of these measurements. Similarly, automatic chambers are also not suitable in this particular design due to the distance between the four treatments replicated in four blocks. Furthermore, to our knowledge, portable cavity-enhanced absorption spectroscopy gas analysers were not available 25 years ago. Instead, the air inside the chamber was sampled from the chamber two to fives time during the measurement (compared to the current 1 Hz frequency) using a syringe, often transferred to pre-evacuated

glass vials, brought back to the laboratory and analysed several days or weeks later by GC-MS. We doubt that measurements could be made every two weeks for a year and half at 96 locations.

We also fully agree that that CH4 flux can be very heterogeneous over time and that low frequency measurements may underestimate annual or average CH4 fluxes. However, if we assumed that this affects all the treatments, the absolute values could be biased but the comparison between treatments remains relevant.

My other key criticism is that the authors need to write more clearly in the introduction about how they investigated the effects of environmental drivers on methanogenesis and methanotrophy (see points 7 and 9). The authors need to revise what they have written or describe the methods they used to infer changes in gross CH4 production or oxidation, given that they did not measure gross methanogenesis or methanotrphy directly.

We confirmed that we did not measure gross methanogenesis and methanotrophy and we therefore revised the manuscript according to points 7 and 9 below.

Specific comments on different sections of the text are provided below.

SPECIFIC COMMENTS

1. Lines 42-43 "...on the CH4 budget of rubber plantation remain poorly understood..." – Please provide references to evidence this claim and to serve as points of comparison with this study.

   In fact, to our knowledge, there has been no previous study on the effect of fertilization on the CH4 budget of rubber plantations. We added a reference on oil palm plantation:

   "While land-use comparisons have been extensively studied, to our knowledge, no previous research has specifically addressed the effect of management practices—particularly fertilization—on the CH4 budget of rubber plantations. A recent study found no effect of reduced fertilization on soil $CH_4$ uptake in an oil palm plantation in Indonesia (Chen et al., 2024)."

2. Line 58 "...CH4 microbial oxidation..." – Word order; "microbial CH4 oxidation" is more grammatically correct.

   Corrected

3. Line 80 "...competes with CH4 for the active site of methane monooxygenase...": Please provide a reference to evidence this claim. This is a well-known phenomena so there are a large range of papers to chose from.

   A reference was provided at the end of the next sentence, but we added two additional ones at the end of this first sentence:

   "A similar substrate competition occurs when NH4+ competes with CH4 for the active site of methane monooxygenase (King and Schnell, 1994; O'Neill and Wilkinson, 1977)"

4. Line 83 "...releasing NH4+ into the soil solution...": Please provide a reference to evidence this claim.

   Reference added (King and Schnell, 1998)

5. Liens 89-90 "...organic substrates derived from primary production, which can be stimulated by fertilizer inputs...": Please provide a slightly more detailed explanation of how fertilizer inputs can stimulate production of organic inputs to soil.

We added more details and the reference suggested at point 22 (Banger et al 2012):

"In addition to anoxic conditions, the main factor controlling methanogenesis is the availability of organic substrates derived from primary production (Liu et al., 2011; Whiting and Chanton, 1993). This availability can increase with fertilizer inputs, due to greater production of above- or below-ground litter (including sloughed-off cells), enhanced decomposition rates, and increased root exudation (Banger et al., 2012; Hobbie, 2005; Melillo et al., 1982; Zhu et al., 2013)"

6. Line 107: Two concerns; first, given that an increasing number of studies are using eddy covariance, automated chambers, and/or modelling to construct trace gas budgets, use of manual flux measurements may not be considered sufficient by some readers. It may be safer to frame the research as process-based CH4 flux study (rather than using the word "budget") or an experiment to establish a preliminary baseline for this type of production system. Second, given that stem fluxes were sampled at only a limited number of timepoints, my recommendation – if you do intend to frame this as CH4 budget study – to frame this as a soil CH4 budget study as I do not think you have enough data for the stem fluxes to do this part of the budget justice.

We understand your concern about using manual flux measurements to and fully agree that the methods we used have limitations and that we have better considered these limitations in the discussion. In addition, we replaced the word "budget" with "uptake" throughout in the text when it was used for our results.

We also agree that the measurement frequency of trunk fluxes was too low to calculate an annual flux. We removed annual trunk flux from Table 2 and instead directly compared soil and trunk fluxes when they were measured together:

"When FT-CH4 was scaled by tree density to allow comparison with FS-CH4, both expressed on a soil surface basis, FT-CH4 offset soil CH4 uptake by less than 0.5% in T1 and T2, and 1.8% in T3 in August 2023. In T4, however, trunk emissions accounted for 3.6% of the combined net CH4 emissions from trunks and soil. In October 2023, FT-CH4 offset soil CH4 uptake by 1.5% in T1 and 14% in T2 and contributed 1.6% in T3 and 0.9% in T4 to the combined net CH4 emissions. In February 2024,the proportion of soil CH4 uptake offset by FT-CH4 was 1.2%, 1.7%, 2.4%, and 4.4% for T1, T2, T3, and T4 respectively."

7. Line 109 "...methanotrophic and methanogenic activities...": Please expand this text to state how you quantified net CH4 flux, methanotrophy and methanogenesis. Quantification of gross methanotrophy or methanogenesis is non-trivial, so experts in the field would be interested to know how you determined the rates of these processes.

We confirm that we did not measure gross methanogenesis and methanotrophy. We replace:

"the drivers of altered methanotrophic and methanogenic activities in the soil" by:

"the factors driving changes in soil CH4 uptake in response to fertilization".

8. Line 122 "...planting density of 500 trees ha−1...": Please indicate if this is a normal planting density for the region, so that non-experts are able to ascertain if this plantation is representative of standard smallholder practices.

   Yes, We mentioned this:

   "…at a planting density of 500 trees ha-1, in accordance with the recommendation of the Rubber Research Institute of Thailand"

9. Line 132 2.2 Methane Flux measurement: The introduction indicates that methanotrophic and methanogenic activity were investigated, but this section does not describe gross $CH_4$ flux measurements or incubation experiments which could be used to directly ascertain methanotrophic and methanogenic activity at different times of year. If direct measurement of methanotrophy or methanogenesis were conducted, then my recommendation is to revise the introduction to better represent the actual research/measurements performed.

   We confirm that we did not measure gross methanogenesis and methanotrophy and revised the introduction as mentioned in our response to point 7 above.

10. Lines 142-143 "Trunk $CH_4$ fluxes (FT-$CH_4$) were measured in August 2023, October 2023, and February 2024 on 8 to 13 trees per treatment.": Since trunk $CH_4$ fluxes were only quantified on 3 occasions, these measurements will provide indicative values but may not enable the researchers to extrapolate more broadly with respect to the annual $CH_4$ budget or overall system behaviour.

    We fully agree that the measurement frequency of stem fluxes was too low to calculate an annual flux. We removed annual trunk flux from Table 2 and instead directly compared soil and trunk fluxes in the result section at the three dates they were measured together, as mentioned in our response to point 6 above.

11. Line 170 "Soil $CH_4$ mole fractions ([$CH_4$]S) were measured at two soil depths (10 and 40 cm) near 24 soil collars (six per fertilization treatments, though not evenly distributed across the four blocks).": Soil gas concentrations appear to have been quantified at the same time as trunk $CH_4$ flux, with 3 sampling campaigns performed over the duration of the study. For clarity, I recommend that the authors state at the start of this section stating that soil $CH_4$ concentrations were only collected 3 times over the course of the study. Again, similar to the point made above (point 8), these measurements will only be indicative.

    We modify the first sentence as suggested:

    "Soil $CH_4$ mole fractions were measured only three times during the study, at two soil depths (10 and 40 cm), near 24 soil collars".

12. Line 190 "Soil mineral nitrogen ($NO_3^-$ and $NH_4^+$) and phosphate ($PO_4^{3-}$) availability was assessed using ion exchange resin bags.": For clarity, add a sentence stating that inorganic N and phosphate were quantified at 4 intervals over the duration of the study. In addition, it's worth stating that some of these measurements coincided with fertiliser application, enabling the investigators to quantify the effects of fertiliser input on inorganic N and P pools. I am aware that you state exactly when these measurements were conducted (lines 197-198), but these revisions at the start of this section would help the reader to understand the overall sampling strategy.

    We added the requested information:

" soil mineral nitrogen (NO3− and NH4+) and phosphate (PO43-) availability was assessed over four periods of 60 to 120 days using ion exchange resin bags"

"May–August 2023 immediately following the first fertilizer application in T2, T3 and T4,

"and October 2023–February 2024 following the second fertilizer application in T3 and T4."

13. Line 250 "...nitrogen concentrations...": The fact that there is no significant impact of fertilizer input rate on soil N concentration implies that the applied N is lost from soil due to plant uptake, N gas flux, and leachate loss.

Probably yes. Moreover, small variations in mineral N concentration might be difficult to detect in the total nitrogen pool

14. Line 264 "3.2 Soil Methane Flux": Given that soil CH4 flux was measured at different distances from the trees (see lines 135-136), was there differences in soil CH4 flux depending on proximity to trees or not? If not, please add a sentence to this section indicating that there did not appear to be differences in soil CH4 at different distances from trees. This is important given that some field studies have identified rhizosphere effects.

The design with three distances from the planting rows was adopted to sample spatial variability related to the planting scheme and fertilizer application, not to investigate a rhizosphere effect. Higher CH4 uptakes were observed at 0.7 m and 3.3 m from the rows than at 2.0 m, especially in the fertilized treatments. We have checked that this was not related to difference in air-filled porosity. This may likely be related to the spatial heterogeneity of fertilizer application. However, it is difficult to discuss a rhizosphere effect because we do not have additional information to support any hypothesis (e.g. fine root biomass, root exudates). We added the following sentence in the results section:

"Lower CH4 uptake was observed at 2.0 m from the planting rows compared to 0.7 m and 3.3 m in the fertilized treatments, likely reflecting spatial heterogeneity in fertilizer application, as fertilizer was broadcast by workers walking approximately 2 m from the planting rows"

15. Figure 2: It would be useful to see these data shown as CH4 mole fractions (ppbv) in addition to the ΔCH4. Please show these data, either by revising Figure 2 or placing the additional data in an appendix.

We agree that it is in fact more useful to show the CH4 mole fractions. We revised Figure 2 as suggested and edit the text to reflect the change in the figure.

16. Line 38 "3.6 Trunk methane flux": One option may be to group the stem flux data with the soil flux and soil concentration data, so that all the CH4 data are grouped together.

We understand your suggestion, but we prefer to keep soil methane fluxes grouped with other soil-related variables (Inorganic nitrogen and phosphorus dynamics and dissolved nitrogen and dissolved organic carbon in lysimeter water).

17. Line 353 "3.7 Methane budget": Given that there are an increasing number of studies that use eddy covariance or automated chamber measurements to quantify CH4

budgets, it may be more appropriate to calculate and discuss the annual CH4 budget as a post-hoc exercise, rather than reporting on the annual budget in the Results; i.e. I recommend that this paragraph is moved to the Discussion and the authors discuss the annual budget in more speculative terms, rather than placing the annual budget calculations in the Results. The reason for this is that manual chamber measurements may not provide high enough frequency measurements to accurately construct an annual budget for CH4, given the high temporal heterogeneity than CH4 is known to exhibit. By presenting these data as a post-hoc calculation in the Discussion, the authors can openly acknowledge potential imperfections in the dataset (e.g. lack of high frequency measurements) while simultaneously providing readers with a first approximation of the what the annual budget is likely to be.

We renamed the last section, previously "Methane budget", as "Annual soil methane uptake", removed annual trunk flux from Table 2, replaced the word "budget" with "uptake" throughout in the text when it was used for our results and better considered limitations related to measurement frequency in the discussion:

"We acknowledge the potential biases associated with interpolating biweekly manual soil flux measurements, particularly given the possibility of high short-term temporal variability. Automated measurements would have been valuable for capturing flux dynamics at finer temporal scales (Barba et al., 2019a; Gana et al., 2018). However, implementing such a system would have been challenging in our experimental plantation, which included four blocks and four fertilizer treatments spread over a 9-ha area, with large distances between chambers and the gas analyzer. Despite these limitations, our findings provide indicative estimates that advance our understanding of the complex interactions between land management practices and greenhouse gas fluxes in tropical agricultural systems"

We also added:

"Therefore, the estimated potential of atmospheric methane removal remains speculative and should be considered as a first approximation to encourage further research in this direction"

However, we think it is better to keep a brief description of these results, with their statistics, in the results section rather than moving them into the discussion.

18. Line 373 "These broad ranges likely reflect differences in edaphic factors across sites...": Can you briefly summarise for the reader what you think the key edaphic differences are which could contribute to variation among sites?

We added what we were thinking about the key edaphic differences which could contribute to variation among sites:

"These broad ranges likely reflect differences in edaphic factors across sites—such as soil texture, porosity, and infiltrability—which influence gas diffusion and soil moisture, and thereby affect CH4 consumption and production."

19. Line 376 "Seasonal variation in FS-CH4 were closely linked to changes in AFP...": My recommendation would be to revise this paragraph so you state from the get go that the findings challenge or are different from prior research, showing that AFP was not as important for modulating CH4 uptake as fertiliser input rates. I would also recommend expanding the discussion to explore why AFP may not be a good predictor in this context; for example, does texture (i.e. sandy loam) or other soil physical properties mean that air/O2 diffuses more readily into the pore spaces? Many

methanotrophs are microaerophilic (i.e. can function at 2% O2 or less), so unless the soil is really pushed close to anaerobiosis then it is possible that they may continue to function effectively even if AFP is low.

We believe the reviewer may have misunderstood our point. AFP exerted strong control over seasonal variation, as expected, and therefore, it does not differ from previous research. Our argument is that the differences between fertilization treatments are not due to differences in AFP. The misunderstanding may stem from the use of the word "variations" instead of "differences", which we have corrected now.

20. Lines 388-391: Variation in the effect of N input on CH4 uptake is likely contingent on background N availability in soil (lines 396-397), with more N-rich soils exhibiting uptake inhibition whereas more N-poor soils are likely to show enhanced CH4 uptake (at least until other resources or environmental conditions constrain methanotrophy). Given that this is something which we have known since the early 2000's (see for example Bodelier et al. 2000 Nature 403), my recommendation would be revise this paragraph to take this knowledge into account as the manuscript would then better acknowledge the theoretical framework that is in place.

The article by Bodelier et al. 2000 published in Nature presents results on rice paddies. These results would be very relevant for studies on wetland or flooded forests but it is not the case for our plantations nor for most rubber plantations in Thailand and Southeast Asia. This is the reason why we were using another article by the same author (Bodelier and Laanbroek, 2004), published in FEMS Microbiology Ecology, which covers more land use types. The conclusion of their section 4 is that "for the time being nitrogen has to be treated as a potentially inhibitory and as a beneficial factor for methane consumption in soils and sediments". We have now recalled their conclusion in our discussion:

"Therefore, nitrogen can potentially inhibit or stimulate CH4 consumption in soils (Bodelier and Laanbroek, 2004)."

21. Lines 405-415: To some extent, some of this information was already raised in the introductory text; there could be value in shortening this paragraph (given that the mechanisms etc. have already been discussed in the intro), to make the text more compact.

We deleted the first sentence that was clearly not needed here and shortened the last one to make the text more compact, as suggested.

22. Lines 416-424: Please include the meta-analysis published by Banger et al. (2012) Global Change Bio as this publication pulls together data from a wide cross-section of papers on fertilizer impacts in rice systems (and is there pertinent within the context of understanding fertilizer impacts on methanogenesis. I believe that they even included data from rice systems that were intermittently drained, which provides insight into the role of moisture dynamics in altering CH4 flux.

Thank you for alerting us about this article. We cited it as a reference on how nitrogen fertilizers stimulate CH4 production:

"Banger et al. (2012) suggested that nitrogen fertilizers may stimulate CH4 production both by alleviating nitrogen limitation to methanogens and by increasing

crop growth, thereby enhancing the availability of carbon substrates for methanogenesis."

We also added the reference of recently published article that shows that a combined nitrogen and phosphorus amendment increased CH4 production in incubated soils from boreal peatland (Byun et al., 2025)

23. Lines 425-434: Joe von Fischer from Colorado State University has also published extensively on microsite-driven production of CH4 using 13C pool dilution, so would recommend reading and referencing his papers here, as there are some interesting insights from his work about C flow via anaerobic pathways (and their implications for CH4 production & emission in otherwise "aerobic" soil).

Thank you also for this suggestion. We found his article published in 2007 in Global Biogeochemical Cycles very useful to improve the discussion on anaerobic microsites in the topsoil:

"Using a isotope-based pool dilution technique, von Fischer and Hedin (2007) demonstrated that small diversions of organic carbon flow from non-methanogenic to methanogenic pathways, likely occurring in anaerobic microsites, can transform soil cores from a net CH4 sink into a net CH4 source. Higher methanogenic activity and greater abundance of Archaea was found in soil cores containing larger amounts of fresh organic matter compared to those with lower amounts when anaerobically incubated (Wachinger et al., 2000)."

24. Lines 435-446: Another, alternative perspective is that it is not the absolute amount of organic matter in the soil that predicts CH4 production but the amount of C flowing through methanogenic pathways (i.e. the "methanogenic fraction", sensu Von Fisher & Hedin 2007). This is a subtle but significant difference, as isotope tracer and pool dilution studies have found it difficult to correlate the absolute amount of organic matter produced by plants (i.e. NPP or inputs of exudates to soil) with methanogenesis. See for instance Von Fisher and Hedin 2007 Global Biogeochemical Cycles and Yang et al. 2017 Global Biogeochemical Cycles 31.

As mentioned above (point 23), we have included the idea that small diversions of organic carbon flow from non-methanogenic to methanogenic pathways, which likely occur in anaerobic microsites, can transform soil cores from a net CH4 sink into a net CH4 source in the discussion. We are sorry but we could not find any article from Yang et al published in 2017 in Global Biogeochemical Cycles.

25. Lines 490 and lines 494-496: Is there published yield response data – i.e. data on the impact of different N input rates on rubber yield for Thailand? Or, are there data on the benefits of alternative N management practices (e.g. use of organic inputs like compost, manure, biochar, etc.). If so, this information could be useful to insert here because to persuade policymakers, local authorities and growers, one needs to provide a credible route towards reducing GHG emissions while also not threatening yields. The statement that "Applying rational fertilization practices to other tree plantations..." implies that growers are currently overfertilizing (which is believable) but data need to be provided showing that lower N inputs could provide similar yield outcomes.

Thank you. We have added this important warning at the end of the discussion.

"However, to convince policy makers, local authorities and producers that implementing rational fertilization practices is a credible pathway to enhance

atmospheric CH4 removal, it is essential to ensure that such practices do not compromise yields and stakeholder's incomes. This was the case for the rubber plantation at our site (Table A7) but remained to be confirmed for rubber plantations in other pedoclimatic context and for other agricultural land-uses"

Yield of latex in our plantation is not affected by the proposed reduction of fertilizer application. We are not aware of other experimental plantations testing fertilisation response of rubber plantation. Cumulative yield over 10 years after the beginning of tapping in our plantation is now included in an appendix (Table A7).

**Review 2**

This study provides a comprehensive investigation of how fertilization rates affect methane emissions from both soil and trunks in rubber plantations. While the dataset is extensive and covers a wide range of indicators, I believe the discussion section could be further refined to enhance the manuscript's clarity and impact.

Thank you for your positive and constructive feedback on our manuscript.

Specific comments

1. Provide experiment design details

- When did fertilization begin on this rubber plantation?

- What was the start date of the treatment?

  We added this information in the revised manuscript:

  "Fertilization treatments began in May 2014, coinciding the start of latex harvesting by tapping.".

- How was the fertilizer applied? Was it evenly distributed across the plantation?

  We added this information in the revised manuscript: "Fertilizer was applied by broadcasting, with workers walking along the interrow at approximately 2 m from the planting rows".

2. Position of soil CH4 flux measurement

In L135-136, soil CH4 fluxes were measured at three distances from the tree rows (0.7, 2.0, and 3.3m).

- Could the authors elaborate on why these particular distances were selected?

  We added this information in the revised manuscript:

  Each plot contained six collars, positioned at three distances from the tree rows (0.7, 2.0, and 3.3 m) to capture spatial variability associated with the planting scheme and fertilizer application

- It would be helpful to discuss whether soil CH4 fluxes vary significantly at different distances.

  We added this information in the revised manuscript:

  "Lower CH4 uptake was observed at 2.0 m from the planting rows compared to 0.7 m and 3.3 m in the fertilized treatments, likely reflecting spatial heterogeneity in fertilizer application, as fertilizer was broadcast by workers walking approximately 2 m from the planting rows".

3. Methane oxidation and production

Since this study only measured the net soil $CH_4$ flux and gradients in soil $CH_4$ mole fractions but did not directly measure the oxidation and production fluxes of soil methane, the discussion of fertilization's specific impacts on these processes should be approached with caution. The discussion that net $CH_4$ production mainly occurred in the top soil layer (L425-434) cannot be definitively supported by the current dataset.

> We agree that our data does not strictly prove that methanogenesis occurs in the top soil layer because we did not incubate soil in anaerobic conditions nor quantify the amounts of methanogenic archaea in the soil, but we still believe that our results strongly support that $CH_4$ production occurs because this is the only plausible explanation for positive $CH_4$ fluxes ($CH_4$ emission from the soil). According to the physic law of gas diffusion, the source must be located closer to the location where the highest concentration is measured than to the location where the lowest concentration is measured. Because higher $CH_4$ concentrations were measured at 10 cm depth that at 40 cm depth, we can reasonably conclude that $CH_4$ production mainly occurred in the top soil layer. We added this argument in the discussion:

> "Interestingly, net $CH_4$ production mainly occurred in the top soil layer in our study. When soil $CH_4$ concentrations exceeded ambient level, they were consistently higher at 10 cm than at 40 cm depth"

4. Contribution of termites and soil invertebrates

L444: "Termite colonies or Scarabaeidae larvae might also contribute to localised hotspots of $CH_4$ production" and L432: "Variations of $O_2$ demand could arise from soil invertebrates, such as leaf-cutting ants and earthworms, that bury plant debris or organic matter". If termites and soil invertebrates are considered important factors influencing $CH_4$ production, please provide more information about their presence and activity in this rubber plantation.

> Termite mounds were present in the plantation as were ant nest, but we did not survey soil invertebrates in this study. We cannot conclude about the importance of soil invertebrates in influencing $CH_4$ production, but we cannot also ignore this possibility in the discussion. We acknowledged this limitation in the revised discussion:

> "Although we did not investigate soil invertebrates in this study, termite mounds and ant nests were present in the plantation."

5. Soil pH impact on methane processes

L410-411" Additionally, the decrease in soil pH observed from T1 to T4 with increased nitrogen addition is another factor known to inhibit soil $CH_4$ oxidation". The significant decrease in soil pH from T1 to T4 is an important finding that warrants more in-depth discussion. Given that soil pH has a profound impact on microbial processes, including methane oxidation and production, please elaborate on the potential mechanisms linking pH changes to methane dynamics.

> We agree, and we deepen the discussion as suggested, for both methanotrophs and methanogens:

> "Although methanotrophs can occur in both acidic and alkaline habitats, they usually grow better at neutral pH (Chowdhury and Dick, 2013; Hanson and Hanson, 1996; Whittenbury et al., 1970; Yao et al., 2023). Liming agricultural soils to raise their pH is known to stimulate soil $CH_4$ oxidation (Abalos et al., 2020; Fonseca de Souza et al., 2025)."

"…suggest that fertilizer application not only suppressed methanotrophic activity but also stimulated methanogenesis, despite concurrent soil acidification. Like methanotrophs, methanogens typically grow better at neutral pH, and methanogenesis has been showed to be limited under low pH conditions in anoxic sediments (Garcia et al., 2000; Phelps and Zeikus, 1984)"

6. Scale extrapolation

The extrapolation of results to a much larger scale in section 4.5 may be too speculative given the significant variability in soil CH4 emission fluxes observed across rubber plantations (as mentioned in L371-372). Additionally, rubber plantations on peatlands may exhibit different responses to fertilization (as mentioned in L422-424), which should be acknowledged when discussing broader implications.

We agree that the extrapolation is speculative. We had already mentioned it before, but we made it stronger now, clarifying that:

"Therefore, the estimated potential of atmospheric methane removal remains speculative and should be considered as a first approximation to encourage further research in this direction"

While rubber plantations established on drained peatlands are not common in Thailand and the Northern part of Southeast Asia, they can be found in Malaysia and Indonesia. So, we also included drained peatlands to the following sentence:

"Specifically, the documented response for the sandy-textured soil at our site may differ from those for soils with higher clay contents, which are expected to exhibit more reductive microsites, or from those of drained peatland"

7. Yield data

It is recommended to supplement the yield data of different treatments. This is of great significance for understanding the treatments described in this article.

Cumulative yield over 10 years after the beginning of tapping is now included in an appendix (Table A7).

**Review 3**

Epron and colleagues presented a very elegant study on methane fluxes from a rubber plantation under different fertilization scenarios, where fertilization increased soil CH4 emissions (or decreased CH4 uptake). The study addresses an interesting topic with potentially impactful results if extrapolated to rubber plantation areas in Southeast Asia. In my opinion, the study was clearly presented, the experimental design was robust, and they put together multiple pieces for successfully understanding patterns and drivers of methane changes: soil fluxes, stem fluxes, soil nutrients dynamics and CH4 concentrations in the soil profile. Soil CH4 fluxes were measured over 37 campaigns for 1.5 years, which was a significant amount of work.

In my opinion, the study presented a couple of limitations. First, one of the two ecosystem components exchanging methane with the atmosphere, tree stems, was poorly measured (just three campaigns). Given the relatively small stem flux rates compared to soil fluxes, this limitation might not be affecting too much the ecosystem upscale results. The second limitation is that the authors discusses methane production and oxidation based on fluxes or processes reflecting net fluxes. Without specific tests on methanogenic or methanotrophic

activities, or without microbial community analysis, I think some part of the discussion might need to be tuned down.

Overall, I think this study is very interesting, well designed and clearly written.

Thank you for your positive and constructive feedback on our manuscript.

We understand your concern about trunk fluxes and agree that the measurement frequency was too low to calculate an accurate annual flux. We removed annual trunk flux from Table 2 and instead directly compared soil and trunk fluxes when they were measured together:

"When FT-CH4 was scaled by tree density to allow comparison with FS-CH4, both expressed on a soil surface basis, FT-CH4 offset soil CH4 uptake by less than 0.5% in T1 and T2, and 1.8% in T3 in August 2023. In T4, however, trunk emissions accounted for 3.6% of the combined net CH4 emissions from trunks and soil. In October 2023, FT-CH4 offset soil CH4 uptake by 1.5% in T1 and 14% in T2 and contributed 1.6% in T3 and 0.9% in T4 to the combined net CH4 emissions. In February 2024,the proportion of soil CH4 uptake offset by FT-CH4 was 1.2%, 1.7%, 2.4%, and 4.4% for T1, T2, T3, and T4 respectively"

We confirm that we did not measure gross methanogenesis and methanotrophy. However, we still believe that our results strongly support that CH4 production and oxidation occur because this is the only plausible explanation for positive (CH4 emission from the soil) and negative (CH4 uptake by the soil) soil CH4 fluxes. Similarly, lower CH4 mole fraction in the soil than in the atmosphere supports the occurrence of CH4 oxidation while higher CH4 mole fraction in the soil than in the atmosphere supports the occurrence of CH4 production. We agree that we cannot conclude that oxidation from 0-10cm were higher than from 10-40 cm and revised the text accordingly (see below).

Minor comments

L125-131. I wonder which the size of the plots was.

We added the missing information:

"The 16 elementary plots (four treatments across four blocks) each contained 108 trees and covered an area of 2160 m²."

L165. Despite the upscaled was explained in Epron et al. 2023a, I think a brief explanation of the upscaling method would be nice here. Did you do it for the whole tree, or just up to the highest measured height?

We added explanation of the upscaling method and remove the reference to Epron et al 2023. We integrated flux between 0 and 3.5 m:

"FT-CH4 values were scaled to the tree level (nmol CH4 s-1 tree-1) by multiplying flux measurements by the corresponding stem surface areas. The trunk of each tree was divided into virtual segments, for which both FT-CH4 and diameter were measured at the chamber location. The length of each virtual segment was calculated as the difference between half the distance to the chamber located above (or 3.5 m height for the upper chamber) and half the distance to the chamber located below (or the height above the ground for the lower chamber). The surface area of each segment was calculated assuming a cylindrical shape and then multiplied by the flux per unit area measured at the corresponding chamber. The integrated fluxes of all trunk segments were summed for each individual tree. Finally, FT-CH4 was multiplied by

tree density to expressed FT-CH4 at the plantation scale, allowing comparison with FS-CH4 on a soil surface area basis."

L235. "with collar identifiers included as a random effect". I assume collars were nested within plots and blocks in the random component of the model as well, right?

Yes, block was initially added in the model but removed as not significant. Plot was not included as there is only one plot per treatment in each block.

L269-273. I do not fully understand the meaning of "median" in these sentences. Does the next suggestion make sense? "The number of measurement days with median positive Fs-CH4 was…"

"For each collar in each treatment, the number of measurement days with positive CH4 fluxes was recorded. This number could range between 0 (all measured fluxes were negative for this collar) to 37 (all measured fluxes were positive for this collar). For the 24 collars in each treatment, both the median and the maximum of the number of days with positive flux were calculated".

We added this information in the subsection "Statistical analyses".

L283. Using "highest" in this sentence is a bit confusing, because T1 is the more negative flux. You could say "average uptake" instead of average Fs-CH4

We updated the sentence following your suggestion:

"The average CH4 uptake was higher in the non-fertilized treatment…"

L359. It seems strange that upscaled trunk emissions were not calculated for each treatment independently, but averaged between trees from different treatments (as shown in Table 2)

Upscaled trunk emissions were not calculated for each treatment because there was no significant difference in trunk CH4 flux between treatment. However, we deleted this annual upscaling from Table 2 because of the too limited number of measurements.

Table 2. I do not fully understand why the authors estimated annual fluxes for two year periods, given that there is an overlap of 6 months between those periods. You cannot really compare between those two periods because it was partially the same data. Additionally, those different estimates are not very discussed in the Discussion section. Please, clarify the rationale.

We agree that it is a pity that we could not measure CH4 fluxes over two years as initially planned because of the Covid 19 pandemics. Unfortunately, we could not calculate annual fluxes for two independent years. Instead, we calculated two annuals fluxes on two periods with 6 months overlapping, covering the second part of the dry season and first part of the rainy season of 2023. The reason is to illustrate variations but the significance of these variations cannot be statistically tested. This is why they are not very discussed. However, we still believe it is interesting to illustrate the variability.

L381. Figure 2 does not seem to fully support this statement. It does not seem that oxidation from 0-10cm were higher than from 10-40cm.

We fully agree and updated the sentence in the discussion:

"The vertical profile of [CH4]S indicated that CH4 oxidation occurred throughout the soil profile, at least down to a depth of 40 cm"

L452. I agree with the authors that a decoupling between soil fluxes and concentrations with stem emissions would suggest internal stem production. However, a decrease of stem fluxes with stem height (as described in the results section) would suggest internal production. A brief discussion about these (apparently) contradictory results might be interesting for the readers.

> We believe that you wanted to write "*However, a decrease of stem fluxes with stem height (as described in the results section) would suggest soil production*". Please note that the vertical patterns were found in most trees but not all, and that the difference between height were not significant. We added the statistical information in the result section:

> "For most trees, FT-CH4 was highest near the base (40–60 cm from the ground) and decreased slightly with height along the trunk, although the differences between heights were not significant (p=0.34)."

> and modify the sentence in the discussion:

> "These findings suggest that CH4 emitted by rubber trees, despite a slight decreasing trend with height along the trunk, may have been produced internally rather than transported from the soil".

Figure A1. I do not really see the value of this figure, but if you want to keep it, consider a colour-blind friendly palette.

> We would like to keep this figure and we changed the colours for light grey and black

---

## Author Response (AR2)

Figure A1 and its caption have been edited using the colour-blind friendly palette "viridis" for R